



# The virtual spot approach: a simple method for image U-Pb carbonate geochronology by high-repetition rate LA-ICP-MS.

Guilhem Hoareau[1], Fanny Claverie[2], Christophe Pecheyran[2], Gaëlle Barbotin[2], Michael Perk[3], Nicolas E. Beaudoin[1], Brice Lacroix[4], E. Troy Rasbury[5]

[1]Universite de Pau et des Pays de l'Adour, E2S UPPA, CNRS, LFCR, UMR5150, Pau, France
[2]Universite de Pau et des Pays de l'Adour, E2S UPPA, CNRS, IPREM, UMR5254, Pau, France
[3]Institute of Operating Systems and Computer Networks Algorithms, TU Braunschweig, 38106 Braunschweig, Germany
4Department of Geology, Kansas State University, 108 Thompson Hall, Manhattan, KS, 66506
[5]Department of Geosciences, FIRST, Stony Brook University, 100 Nichols Rd, Stony Brook, NY 11794-2100, USA

*Correspondence to*: Guilhem Hoareau (guilhem.hoareau@univ-pau.fr)

**Abstract.** We present a simple approach to laser ablation inductively coupled plasma mass spectrometry (LA-ICP-MS) U-Pb dating of carbonate minerals from isotopic image maps, made possible using a high repetition rate femtosecond laser ablation system. The isotopic ratio maps are built from 25-μm-height linear scans, at a minimal repetition rate of 100 Hz. The analysis

of $^{238}U$, $^{232}Th$, $^{208}Pb$, $^{207}Pb$ and $^{206}Pb$ masses by a sector field ICP-MS is set to maximize the number of mass sweeps, and thus of pixels on the produced image maps (∼8 to 19 scans s$^{-1}$). After normalization by sample standard bracketing using the Iolite 4 software, the isotopic image maps are discretized into squares. The squares correspond to virtual spots of chosen dimension, for which the mean and its uncertainty are calculated, allowing to plot corresponding concordia diagrams commonly used to obtain an absolute age. Because the ratios can vary strongly at the pixel scale, the values obtained from the virtual spots display

higher uncertainties compared to static spots of similar size. However, their size, and thus the number of virtual spots can be easily adapted. A low size will result in higher uncertainty of individual spots, but their higher number and potentially larger spread along the isochron can result in a more precise age. Reliability of this approach is improved by using a mobile grid for the virtual spot dataset of a set size, returning numerous concordia allowing to select the more statistically robust result. One can also select from all the possible spot locations on the image map, those that will enable regression to be obtained with the

best goodness of fit. We present examples of the virtual spot approach, for which in the most favorable cases (U > 1 ppm, $^{238}U/^{206}Pb \gg 1$, and highly variable U/Pb ratios) a valid age can be obtained within reasonable uncertainty (< 5-10%) from maps as small as 100 μm x 100 μm, i.e. the size of a single spot in common in situ approaches. Although the method has been developed on carbonates, it should be applicable to other minerals suited to U-Pb geochronology.



## 1 Introduction

The in-situ uranium-lead (U-Pb) dating of carbonate minerals (calcite, dolomite) by laser ablation inductively coupled plasma mass spectrometry (LA-ICP-MS) is now a well-established approach (Roberts et al., 2020). Due to the ubiquitous nature of carbonates in the upper crust, and to the sub-millimetre-scale spatial resolution of the method, in situ U-Pb dating has been applied successfully to a variety of geological contexts and objects such as tectonic fractures and veins (e.g. Beaudoin et al.,

2018; Nuriel et al., 2019; Roberts and Holdsworth, 2022), carbonate deposition (e.g. Drost et al., 2018; Montano et al., 2021), speleothems (e.g. Woodhead and Petrus, 2019) or cements (e.g. Brigaud et al., 2020; Motte et al., 2021). Aside from the LA-ICP-MS approach used by most laboratories worldwide, which consists in the construction of isochrons from the combination of several tens of ablation craters (80 to 235 µm) made on the same or adjacent crystals, recent studies have demonstrated the feasibility of obtaining carbonate U-Pb ages from isotope ratio maps (Drost et al., 2018; Roberts et al., 2020; Hoareau et al.,

2021; Rochín-Bañaga et al., 2021; Davis and Rochín-Bañaga, 2021; Liu et al., 2023). This approach has also successfully been applied to zircon (Chew et al., 2017), apatite (Ansberque et al., 2020; Liu et al., 2023) and monazite (Chew et al., 2021). Its principle is identical to that used to make elemental mineral concentration maps by LA-ICP-MS, i.e. rastering the laser spot along successive (usually horizontal) lines by moving the stage, combined with continuous isotopic mass measurement of the ablation products (e.g., Kosler, 2008). The time-resolved signals obtained for each line are then combined to form the image

maps, so as each pixel corresponds to a mass sweep of the spectrometer. An obvious advantage of this approach is that it allows both to visualize the distribution of trace elements over the analyzed area, and to obtain an age in the most favourable cases (U and Pb contents typically above 1 ppm, and variable, positive U/Pb ratios) (Drost et al., 2018). As with concentration maps, the analysis of several element masses (in addition to those useful for dating) can identify areas that may correspond to solid inclusions (e.g. clays) or to diagenetic alteration (e.g., Roberts et al., 2020). Filtering-out the corresponding pixels makes it

possible to keep only the most favorable zones for dating, and thus maximize the chances of obtaining a reliable age from the masses used for U-Pb dating (typically $^{238}U$, $^{207}Pb$ and $^{206}Pb$) (Drost et al., 2018; Roberts et al., 2020; Hoareau et al., 2021). Whereas the analytical conditions used for image-based carbonate dating are quite comparable to those used for "traditional" elemental mapping, the data treatments necessary to calculate an age from isotopic image maps are highly variable among studies published so far, with possible repercussions on the reliability of the ages obtained. The study of Drost et al (2018)

showed the potential of the pixel pooling approach, which uses ratios other than those used for dating ($^{238}U/^{208}Pb$, $^{207}U/^{235}U$) to sort the pixel values, split the resulting empirical cumulative distribution function (ECDF) into subsets, and calculate mean ratios and their uncertainty for each subset ("pseudo-ellipses"). This method tends to maximize the spread of the subset ratios along an isochron, ideally resulting in more precise ages. The potential for obtaining accurate and very precise ages even with a quadrupole ICP-MS has also been highlighted by Roberts et al. (2020) and Hoareau et al. (2021). However, as pointed out

by the authors, the sorting approach of Drost et al. (2018) cannot be used blindly as it is likely to cause biases in the calculated ages (i.e., precise but inaccurate ages) if the sample is not well characterized and the pixel values filtered adequately. For example, using either $^{238}U/^{208}Pb$ or $^{207}U/^{235}U$ as the sorting ratio may result in two distinct, but statistically plausible ages (i.e.,





MSWD close to 1). In addition, the calculated common Pb values may be biased (Hoareau et al., 2021). In the latter study, we also presented an alternative approach consisting in running a robust regression through the pixel values in a concordia

diagram. In most cases, it allows to obtain ages identical to those of Drost et al. (2018) both in terms of accuracy and precision. However, the approach of Drost et al. (2018) shows better performance in terms of accuracy when the U concentrations of the analyzed samples are low (typically a few hundred ppb). Finally, Davis and Rochín-Bañaga (2021) and Liu et al. (2023) have recently presented another approach, based on the use of Bayesian inference in age calculation. In this approach, it is first better to calculate an age and common Pb range by a classical regression approach (York type) through the pixels to which

uncertainties related to the number of counts have been added. Then, a planar regression in a 3D concentration diagram is used to refine this age by Bayesian statistics. This approach shows great potential and will likely gain major attention in future studies. However, it is likely that the bias in age results reported in some cases by Hoareau et al. (2021) and in the present study may also apply with this approach, since pixels located far from the discordia line in a concordia diagram, which do not necessarily correspond to clear outliers in time-resolved data, are also considered in the Bayesian regression.

In their image-based dating methodology, Hoareau et al (2021) used a high repetition rate femtosecond laser (500 Hz), allowing to use a small spot width (15 µm), coupled to a high-resolution SF-ICP-MS. Both allowed to obtain highly spatially resolved image maps (25 µm rasters) and with a good analytical sensitivity. In addition to the robust regression method that is the main focus of their work, Hoareau et al (2021) also presented an approach based on a squaring of the image maps, with averaging and uncertainty calculation for each square called "pseudo spot". It was intended to check the accuracy of the ages obtained

by the other methods compared (robust regression, and the method of Drost et al. 2018). In the Hoareau et al. study (2021), this discretization was performed after averaging the number of pixels, which results in uncertainties too high for the obtained ages to be satisfactory for use in geoscience case studies. In the present study, we further develop the image discretization (squaring) approach. On the one hand, we avoid, or limit averaging the number of pixels to maximize the spatial resolution of the image maps and improve the statistics of the calculated ratios. On the other hand, we propose evolutions allowing to

calculate several ages for a single image map, either by moving the grid allowing its discretization into virtual spots, or by creating sub-images within the image. In the latter case, a weighted average of ages obtained at different locations in the image map can be calculated. We show that this approach, which we call here "virtual spots", is well suited to highly spatially resolved images. It can be used to obtain ages comparable to those calculated by classical approaches based on in situ spots, while being flexible and simple in its implementation.

**2 Samples**

Seven (7) samples of carbonates have been chosen to test the new approach, among which 5 have been previously dated. Among the 7 samples, two (BH14 and C6-265-D5) were previously analyzed as part of the study of Hoareau et al. (2021). The samples are:



(i) a tectonic calcite vein (AUG-B6) from the Paris basin (France) dated by U–Pb LA-ICP-MS spot analyses (i.e. range of spot ablations) at about 42 to 43 Ma in several laboratories (Pagel et al., 2018; Blaise et al., 2023), including ours (42.8 ± 2.0 Ma (2s); MSWD = 3.7; the detailed methodology is presented in the Supplement S1 including Table S1 and Fig. S1). A petrographical description of AUG-B6 is available in Pagel et al. (2018).

(ii) a tectonic calcite vein (BH14) from the Bighorn Basin (Wyoming, USA) dated by U–Pb LA-ICP-MS spot analyses (i.e. range of spot ablations) at 63.0 ± 2.2 Ma (MSWD=1.6) by Beaudoin et al. (2018) and at 61.2 ± 2.9 Ma (MSWD = 4.1) in our laboratory (see Hoareau et al., 2021).

(iii) a dolomite cement (C6-265-D5) found in a tectonic breccia affecting Callovo-oxfordian limestones of the northern Pyrenees (France). It was dated at 106.1 ± 5.5 Ma from U-Pb LA-ICP-MS by the image-based method of Drost et al. (2018) but using WC1 as the primary standard (see Motte et al., 2021, including a petrographical description of the sample as DC4$_{Meillon}$).

(iv) a lacustrine limestone (Long Point; Duff Brown Tank locality in the Colorado Plateau, USA), precisely dated by Hill et al. (2016) at 64.0 ± 0.7 Ma (2s) by U–Pb methods using isotope dilution (ID) multi-collector inductively coupled plasma mass spectrometry (MC-ICP-MS), and labelled DBT in the following. This sample is widely used as a validation RM.

(v) a Tithonian dolostone from the northern Pyrenees (Senz7) which we have precisely dated by ID-MC-ICP-MS at 147.0 ± 2.4 Ma (2s) (see Supplement S2 including Fig. S2 for the detailed methodology). Like C6-265-D5, this sample was also dated in April 2019 to ~137 Ma (~7% too young) by the image-based method of Drost et al. (2018), using WC1 as the primary standard (see Motte et al., 2021, including a petrographical description of the sample as RD1$_{Mano}$).

(vi) a calcite-cemented sedimentary breccia (Collings Ranch Conglomerate) from the Arbuckle Mountain (USA). The calcite cement (ARB20-2D) has not been previously dated. The intergranular cement is made of blocky calcite, dull in cathodoluminescence excepted rare grains with concentric zoning defining a second calcite generation (Fig. S3).

(vii) a deformation band affecting a calcarenite (Cot2a) from the Cotiella basin of Cretaceous age in southern Pyrenees, Spain (see Taxopoulou et al., 2023 for a petrographical description). The calcite cements located in the deformation band have not been previously dated.

## 3 Analytical strategy

All the samples were analyzed with a 257 nm femtosecond laser ablation system (Lambda3, NEXEYA, Bordeaux, France) coupled to a sector field inductively coupled plasma mass spectrometer (SF-ICP-MS) Element XR (Thermo Fisher Scientific, Bremen, Germany) fitted with the Jet Interface, at the IPREM laboratory (Université de Pau et des Pays de l'Adour, Pau, France), in October 2018 (BH14), April 2019 (C6-265-D5), April 2022 (AUG-B6, ARB20-2D, DBT), and May 2023 (Senz7, Cot2a). The analytical conditions are essentially like those previously detailed in Hoareau et al. (2021). Before 2020, polished chips were ablated at a repetition rate of 500 Hz with a fluence of ~2 μJ per pulse, along 23 to 25 linear scans of 0.72 to 1.21 mm width (Table 1). These lines are 25 μm height, separated by 25 μm so that they are adjacent to each other, were obtained



using a back-and-forth movement of the laser (at 5 mm s$^{-1}$) combined with a stage movement rate of 25 µm s$^{-1}$. They correspond to 29 to 48.5 s of analysis per linear scan, followed by a 15 s break. After 2020, polished chips or sections (30 or 80 µm-thick) were ablated either at 100 Hz or 500 Hz (May 2023), a fluence of ~7 µJ per pulse, along 8 to 32 linear scans of 0.96 to 3.62 mm width. The adjacent lines are 25 µm height, with similar laser and stage movement rates than previous sessions, except

that breaks between lines were increased to 25 s. Ablations correspond to 38.3 to 145 s of analysis per linear scan. All sessions considered, the total analysis time was of ~5.5 to 31.3 min for a complete image map of a surface of between 0.45 and 1.16 mm$^2$ (Table 1). The ablation depth is about 25 µm at 100 Hz, and 40 µm at 500 Hz as measured with a digital microscope. Before analysis, all samples were pre-cleaned with the laser using a stage movement rate of 200 µm s$^{-1}$.

| Sample name | Date | Line width (mm) | Ablation duration per line (s) | Number of lines | Total analysis time (min) | Number of pixels | Area (mm$^2$) | Repetition rate (Hz) | Dwell time (ms) |
|---|---|---|---|---|---|---|---|---|---|
| **ARB20-2D** | 03/2022 | 3.625 | 145 | 8 | 19.7 | 20488 | 0.725 | 100 | 57 |
| **AUG-B6** | 03/2022 | 0.957 | 38.3 | 8 | 5.52 | 5208 | 0.191 | 100 | 57 |
| **BH14** | 10/2018 | 1.212 | 48.5 | 23 | 18.8 | 8326 | 0.697 | 500 | 134 |
| **C6-265-D5** | 04/2019 | 0.722 | 28.9 | 25 | 12.3 | 10675 | 0.452 | 500 | 68 |
| **Cot2a** | 05/2023 | 1.450 | 58.0 | 32 | 31.3 | 33696 | 1.16 | 500 | 57 |
| **DBT** | 03/2022 | 1.950 | 78.0 | 8 | 10.8 | 11024 | 0.390 | 100 | 57 |
| **Senz7** | 05/2023 | 1.237 | 49.5 | 8 | 7.02 | 7000 | 0.248 | 100 | 57 |
| **WC / NIST** | | 0.550 | 22 | 8 | 3.35 | ~3100 | 0.11 | 100 / 500 | 57 / 68/ 134 |

**Table 1: Operating conditions**

The aerosol generated by ablation was transported to the ICP-MS using a helium (He) stream at 600 mL min$^{-1}$, except in October 2018 (BH14) when argon (Ar) was used at 650-700 mL min$^{-1}$. The washout time for the ablation cell was approximately 500-600 ms for He gas and ~1 s for Ar gas, based on the 99% criterion. To enhance sensitivity, 10 mL min$^{-1}$ of nitrogen was added to the twister spray chamber of the ICP-MS through a tangential inlet, while He was introduced via another

tangential inlet located at the top of the spray chamber. All measurements were performed under dry plasma conditions. The femtosecond laser ablation inductively coupled plasma mass spectrometry (fs-LA-ICPMS) setup was tuned daily to optimize sensitivity, accuracy, particle atomization efficiency, and stability. The additional Ar carrier gas flow rate, torch position, and power were adjusted to achieve a U/Th ratio close to 1±0.05 during the ablation of NIST SRM612 glass. Daily checks were performed for detector cross-calibration and mass bias calibration using the Element software sequence. The laser and ICP-

MS parameters for U–Pb dating are detailed in Table A1. The selected isotopes were $^{238}$U, $^{232}$Th, $^{208}$Pb, $^{207}$Pb, and $^{206}$Pb,





resulting in a total mass sweep time of ~57 ms (except for ~139 ms in October 2018 for BH14 and ~68 ms in April 2019 for C6-265-D5). As of May 2023 (500 Hz), the detection limits were approximately 1.1 ppb for $^{206}$Pb and 0.02 ppb for $^{238}$U, while the quantification limits were about 3.5 ppb for $^{206}$Pb and 0.07 ppb for $^{238}$U. The unknown samples were bracketed with NIST SRM612 (before 2020: BH14, C6-265-D5) and 614 (after 2020: ARB20-2D, AUG-B6, Cot2a, DBT, Senz7), as well as the
commonly used WC-1 calcite RM (Roberts et al., 2017). Small maps of the primary RMs (~0.1 mm², ~3.5 min of analysis time) were generated before and after each unknown sample analysis under similar conditions. Details on the laser and ICP-MS parameters used for U–Pb dating can be found in Appendix A.

## 4 Data processing

### 4.1 Initial data processing in Iolite

U-Pb data were processed using Iolite 4 software (Paton et al., 2011) and the VizualAge_UcomPbine Data Reduction Scheme for background correction and normalization (Chew et al. 2014). After line selection and background correction, NIST SRM614 glass was used as the primary reference material for normalization (mass drift and interelement fractionation) of both Pb/Pb and Pb/U isotope data. Pb/Pb ratios are taken from Woodhead and Hergt (2001), while the $^{206}$Pb/$^{238}$U ratio (0.80612) was calculated from Woodhead and Hergt (2001), Duffin et al. (2013), Jochum et al. (2011), and CIAAW-IUPAC. Correction
of additional matrix-related offset of the $^{206}$Pb/$^{238}$U ratio used WC-1 calcite reference material (age 254.4 ± 6.4 Ma), using the method of Roberts et al. (2017). The DBT limestone (Age 64.04 ± 0.67; Hill et al., 2016) was used as validation reference material. The ratio values for each pixel are obtained in the "Imaging" section of Iolite. The software allows the operator to filter out pixels that are considered anomalous, like the Monocle plugin of the previous Iolite version. Here, only pixels with negative ratio values were excluded, except for sample AUG-B6. For the latter, the presence of numerous spikes on the
$^{207}$Pb/$^{206}$Pb ratios result in expected age values but very high common lead values (0.85-0.9) in Tera-Wasserburg (TW) diagrams. To obtain values closer to those expected (0.8-0.85), pixels with $^{207}$Pb/$^{206}$Pb values higher than 1.5 were removed.

### 4.2 Python API processing

An in-house Python script was then used as part of the Python API embedded in Iolite 4. This script as well as the ones described below are publicly available (Hoareau et al., 2024). The Python script allows to reconstruct isotopic ratio matrices
from pixel values, to add excess variance to individual ratio uncertainties, to correct of matrix-related offset of the $^{206}$Pb/$^{238}$U ratio, to split the isotopic image maps into virtual spots forming a grid, and to calculate the mean and its uncertainty for each spot, for all ratios. First, for each virtual spot the mean of ratios is calculated, pixels identified as outliers, if any, are removed (i.e, pixel ratio values outside the 95% confidence interval of the standard error), and the mean recalculated. Second, excess variance calculated by Iolite 4 using all ablation lines of NIST 614 (typically 1.5-2.5% (2s) for $^{238}$U/$^{206}$Pb and 0.1-0.3 (2s) for
$^{207}$Pb/$^{206}$Pb) is added by quadrature to the uncertainties of each virtual spot obtained from the unknowns (and WC-1) within the session. For correction of the matrix-related offset, the isotopic image of WC-1 produced from all lines obtained in the





analytical session is split in virtual spots of size like those used on the unknowns. The mean and uncertainty values of $^{238}U/^{206}Pb$ and $^{207}Pb/^{206}Pb$ ratios are then plotted in TW diagrams using IsoplotR (Vermeesch, 2018) to calculate its age used for correction. After these first steps, virtual spots are calculated for the unknowns. Whereas their minimum height (vertical size) is limited to that a line (25 µm), their minimum width (horizontal size) can theoretically be as small as that of a few microns (few pixels). In that case, the number of virtual spots can exceed several hundreds, resulting in unreasonably high computing times when it comes to age calculation using IsoplotR. In that case, Isoplot (Ludwig, 2003) can be used instead. The script also allows to displace the grid on the matrix, so that new spatial distributions of the virtual spots are obtained (Fig. 1). To achieve this, it may be necessary to adjust the size of the virtual spots very slightly (e.g., 51 µm x 50 µm instead of 50 µm x 50 µm) to ensure an integer number of pixels per spot. Finally, virtual spots with high individual uncertainties can be filtered out.

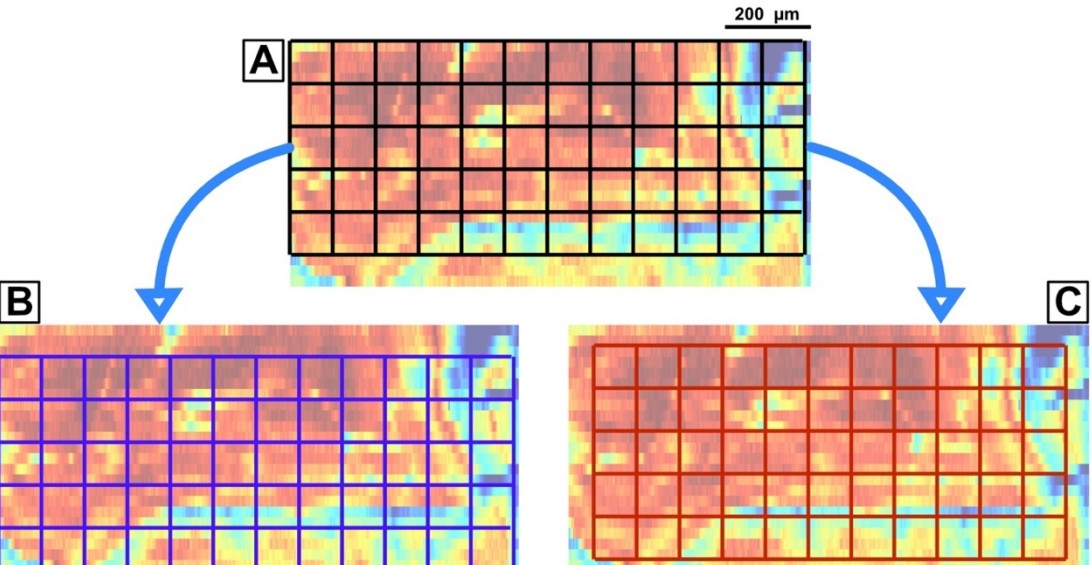

**Figure 1: Principle of image map discretization (based on BH14). A: In its default position, the grid defines 60 squares (corresponding to 60 virtual spots). The bottom and the right part of the image map are not considered, as the squares are not complete. B: Example of another grid position, still defining 60 squares. C: Another position where more squares are incomplete and therefore ignored, reducing the total number of squares to 55.**

### 4.3 Age calculation using Python / R

Ages are calculated either from $^{238}U/^{206}Pb$ and $^{207}Pb/^{206}Pb$ (TW) or $^{206}Pb/^{238}U$ and $^{207}Pb/^{235}Pb$ ratios (concordia), and corresponding diagrams generated with an R script using the *age* and *concordia* functions of IsoplotR library. Systematic uncertainties are then added quadratically to the final age. They comprise the decay constant uncertainty of $^{238}U$ (0.1 %, 2 s), the $^{238}U/^{206}Pb$ ratio uncertainty of WC1, as estimated by Roberts et al. (2017) (2.7 %, 2 s), and the long-term excess variance taken as 2.0 % (2 s). Three types of processing are proposed to obtain ages from the image maps:



(i) First, for a single sample, the ability to change the grid location makes it possible to calculate several dozens of ages corresponding to each grid location ('mobile grid' method). This process allows to assess the homogeneity of the sample in terms of age by using weighted mean statistics, and to select the best age obtained in terms of precision and statistical robustness (MSWD value and p).

(ii) A second algorithm has been developed to calculate the best possible regression in terms of statistical robustness. An orthogonal regression is first performed in a concordia diagram using the values obtained for all the virtual spots that can be defined on the image map (several thousand). The spots can therefore be largely overlapping in space. For a good quality sample, all the values are expected to define a robust linear trend in the concordia diagram, defined by MSWD values close to unity. The regression uses Scipy's ODR function, whose slope and intercept results are strictly identical to a York-type regression. If the age is like those obtained by the first method (mobile grid), the $n$ points with a minimum orthogonal distance to the regression are selected (n = 1000 for example). The corresponding virtual spots can still largely overlap. Finding the largest set of non-overlapping rectangles can be formulated as an independent set problem that uses Boolean variables $x_i$ for every rectangle (Equation 1):

$$max \sum_{x_i} x_i \qquad (1)$$

$$\text{s. t.} \quad \overline{x_i} \vee \overline{x_j} \quad \forall \text{ intersecting } rectangles \text{ i, j}$$

$$x_i \in \{0, 1\}$$

The constraint programming solver (CP-SAT) is used to solve the model and find the maximum number of non-overlapping virtual spots in the image map. A unique age is then calculated using IsoplotR. Very good goodness-of-fit (GoF) parameters (MSWD, p) are expected. The method is labelled 'Rectis' method in the following.

(iii) The last processing method ('sub-image' method) can calculate a set of ages obtained from creating sub-images within the isotopic image map. In detail, virtual spots of small size (for example, 25 µm x 25 µm) are first created in Iolite. Then, the isotopic image is splitted in sub-images of chosen dimension (for example 100 µm x 100 µm). For each sub-image, an age is calculated from the virtual spots it contains (i.e., 16 spots of 25 µm x 25 µm for a 100 µm x 100 µm image, or 64 spots for a 200 µm x 200 µm image). As presented in the following, provided samples are suitable, this approach allows to calculate a weighted average of ages obtained at different locations in the image map. It also theoretically makes it possible to obtain reliable ages from image maps of very limited area, and with extremely short analysis times (~ 1.3 to 3 min without the standards).

## 5 Results

The mean U, Pb, and Th concentrations of studied samples, and their $^{238}$U/$^{206}$Pb and $^{207}$Pb/$^{206}$Pb ratios, are summarized in Table 2. Corresponding image maps are presented in Fig. 2.





| Sample name | Reference age (Ma) | U (ppm) | Unc. (2SD) | Pb (ppm) | Unc. (2SD) | Th (ppm) | Unc. (2SD) | $^{238}U/^{206}Pb$ | Unc. (2SD) | $^{207}Pb/^{206}Pb$ | Unc. (2SD) |
|---|---|---|---|---|---|---|---|---|---|---|---|
| ARB20-2D | NA | 0.733 | 0.075 | 0.22 | 0.05 | 0.0011 | 0.0010 | 6.71 | 0.75 | 0.558 | 0.031 |
| AUG-B6 | ~42.5 ± 1.0[1] | 4.70 | 1.50 | 0.30 | 0.29 | 0.028 | 0.069 | 43.3 | 16.5 | 0.66 | 0.10 |
| BH14 | 63.0 ± 2.2[2] | 10.7 | 9.4 | 0.27 | 0.10 | 0.008 | 0.017 | 51.4 | 26.1 | 0.40 | 0.17 |
| C6-265-D5 | ~106.1 ± 5.5[3,*] | 0.28 | 0.22 | 0.059 | 0.045 | 0.064 | 0.051 | 19.2 | 40.5 | 0.60 | 0.14 |
| Cot2a | NA | 1.27 | 0.93 | 14 | 13 | 2.4 | 1.8 | 1.01 | 1.11 | 0.832 | 0.021 |
| DBT | 64.0 ± 0.7[4] | 27 | 3.8 | 0.81 | 0.63 | 0.0295 | 0.0055 | 46.5 | 15 | 0.40 | 0.15 |
| Senz7 | 147.0 ± 2.4[5] | 6.5 | 1.6 | 0.22 | 0.12 | 0.338 | 0.048 | 31.3 | 3 | 0.256 | 0.042 |

**Table 2: Mean U, Pb, Th concentrations, and $^{238}U/^{206}Pb$ and $^{207}Pb/^{206}Pb$ ratios of studied samples, as calculated by Iolite4 from raster lines. [1]Pagel et al. (2018) and Blaise et al. (2023). [2]Beaudoin et al. (2018), [3]Motte et al. (2021), [4]Hill et al. (2016), [5]This study. \*Not corrected from bias due to the use of a calcite primary standard.**





**Figure 2: LA-ICP-MS image maps of U, Pb, Th concentrations (in ppm), and $^{238}U/^{206}Pb$ and $^{207}Pb/^{206}Pb$ ratios of studied samples. All maps are at the same scale. Note that concentrations are estimated from NIST SRM, and $^{238}U/^{206}Pb$ ratios are not corrected from carbonate RMs.**





## 5.1 Examples of ages calculated with mobile grids of different virtual spot sizes

### 5.1.1 Case of high-U samples

We present ages calculated with different virtual spot sizes for ARB20-2D, AUG-B6, BH14, DBT, and Senz7, which gave satisfactory results owing to high U and Pb contents. For samples of age already determined by other studies (AUG-B6, BH14, DBT, Senz7), the age and common Pb values obtained here are identical to the reference ones within uncertainties, whatever the size of the virtual spots chosen (Table 3; Fig. 3). Moving the grid over the image maps enables to select the best results from among the different ages calculated for a given spot size (Table 3; Fig. 3). For the previously undated sample ARB20-

2D, age and common Pb values of ~305-320 Ma and ~0.82 are obtained, respectively. A Pennsylvanian age is fully consistent with the inferred age of deposition of the host conglomerates (Ham, 1954), suggesting their very early cementation. These satisfactory results agree with the good linear distribution of pixel values in a Tera-Wasserburg plot for most samples (Fig. S4). When the default grid is selected (i.e., covering the entire image map surface and therefore with a maximum number of virtual spots), age uncertainty decreases with decreasing virtual spot size (Table 3). This is due to the larger number of virtual

spots and their larger spread along the isochron, despite a larger individual uncertainty for each spot, as detailed by Kylander-Klark (2020) and Roberts et al. (2020) based on conventional spot analyses. Selecting the best results puts an end to such correlation, as low age uncertainties can be obtained even for a small number of spots. In the extreme case of spots as low as 25 µm x 25 µm, the resulting age uncertainty can be below 2% (without propagation of external uncertainty). Most samples also display MSWD values higher than 1 that increase with the size of the virtual spots, due to lower individual uncertainties.

On the one hand, these values indicate some heterogeneity of the samples, also visible on the TW diagrams (Figs 3 and 4). Such heterogeneity is, for example, also visible in the conventional spot analyses carried out in the laboratory for samples BH14 and AUG-B6 (Fig. S1 and Fig. 3 of Hoareau et al. (2021)). On the other hand, they are increased by the large error correlation between ratios that the virtual spot approach generates for the most favourable samples (here BH14 and ARB20-2D), as discussed in part 6.1.


| Sample name | Virtual spot size (µm) | Grid in its default location (maximum number of virtual spots) | | | | Best results selected from multiple grid locations | | | |
|---|---|---|---|---|---|---|---|---|---|
| | | Number of virtual spots | Age (without/with systematic uncertainty) (Ma) | Pb0 | MSWD | Number of virtual spots | Age (without/with systematic uncertainty) (Ma) | Pb0 | MSWD |
| **ARB20-2D** | 200 x 200 | 18 | 314.7 ± 15.8 / 18.8 | 0.815 ± 0.0190 | 4.8 | 17 | 319.9 ± 12.3 / 16.2 | 0.816 ± 0.0147 | 2.4 |





| | | | | | | | | | |
|---|---|---|---|---|---|---|---|---|---|
| | 150 x 150 | 24 | 317.9 ± 16.0 / 19.1 | 0.818 ± 0.0211 | 4.3 | 23 | 318.3 ± 14.1 / 17.5 | 0.820 ± 0.0180 | 3.5 |
| | 100 x 100 | 72 | 312.4 ± 7.5 / 12.7 | 0.813 ± 0.0101 | 2.7 | 70 | 313.1 ± 6.7 / 12.3 | 0.813 ± 0.0088 | 1.9 |
| | 75 x 75 | 94 | 310.3 ± 7.4 / 12.6 | 0.809 ± 0.0106 | 2.4 | 92 | 311.8 ± 6.2 / 12.0 | 0.813 ± 0.0088 | 1.8 |
| | 50 x 50 | 284 | 311.4 ± 4.1 / 11.0 | 0.816 ± 0.0062 | 1.4 | 284 | 311.4 ± 4.1 / 11.0 | 0.816 ± 0.0062 | 1.4 |
| | 25 x 25 | 1136 | 305.7 ± 3.3 / 10.8 | 0.817 ± 0.0054 | 1.4 | | | | |
| **AUG-B6** | 150 x 150 | 6 | 45.5 ± 3.2 / 3.5 | 0.865 ± 0.026 | 0.6 | 6 | 45.5 ± 3.2 / 3.5 | 0.865 ± 0.026 | 0.6 |
| | 100 x 100 | 18 | 44.2 ± 2.2 / 2.6 | 0.861 ± 0.019 | 1.1 | 18 | 44.2 ± 2.2 / 2.6 | 0.861 ± 0.019 | 1.1 |
| | 75 x 75 | 24 | 41.2 ± 2.8 / 3.1 | 0.842 ± 0.024 | 1.7 | 11 | 42.4 ± 2.6 / 3.0 | 0.872 ± 0.022 | 0.94 |
| | 50 x 50 | 72 | 42.0 ± 2.1 / 2.5 | 0.853 ± 0.020 | 1.7 | 54 | 42.4 ± 1.7 / 2.2 | 0.860 ± 0.017 | 1.2 |
| | 25 x 25 | 296 | 41.1 ± 1.5 / 2.0 | 0.859 ± 0.016 | 1.4 | | | | |
| **BH14** | 200 x 200 | 12 | 61.8 ± 1.7 / 2.7 | 0.735 ± 0.0203 | 8.6 | 5 | 61.4 ± 0.5 / 2.1 | 0.727 ± 0.0069 | 1.1 |
| | 150 x 150 | 24 | 62.1 ± 1.0 / 2.3 | 0.732 ± 0.0124 | 5.3 | 14 | 62.0 ± 0.4 / 2.1 | 0.729 | 1.2 |
| | 100 x 100 | 60 | 62.5 ± 0.7 / 2.2 | 0.734 ± 0.0086 | 4.7 | 44 | 61.6 ± 0.6 / 2.1 | 0.722 ± 0.0073 | 2.4 |
| | 75 x 75 | 112 | 62.0 ± 0.6 / 2.2 | 0.730 ± 0.0071 | 4.6 | 90 | 62.1 ± 0.5 / 2.1 | 0.729 ± 0.006 | 2.5 |
| | 50 x 50 | 242 | 62.4 ± 0.5 / 2.2 | 0.738 ± 0.0054 | 3.6 | 230 | 62.1 ± 0.4 / 2.1 | 0.732 ± 0.0051 | 2.9 |
| | 25 x 25 | 1173 | 62.3 ± 0.3 / 2.1 | 0.738 ± 0.0034 | 2.1 | | | | |
| **C6-265-D5** | 200 x 200 | 9 | 82.0 ± 25.0 / 25.1 | 0.781 ± 0.034 | 2.4 | 6 | 45.0 ± 15.0 / 15.1 | 0.722 ± 0.029 | 2.0 |
| | 150 x 150 | 16 | 68.0 ± 30.0 / 30.1 | 0.745 ± 0.024 | 6.6 | 9 | 57.0 ± 15.0 / 15.1 | 0.743 ± 0.031 | 2.0 |



| | | | | | | | | | |
|---|---|---|---|---|---|---|---|---|---|
| | 100 x 100 | 42 | 76.3 ± 17.1 / 17.3 | 0.760 ± 0.020 | 3.7 | 30 | 66.1 ± 13.1 / 13.3 | 0.748 ± 0.026 | 1.6 |
| | 75 x 75 | 72 | 75.0 ± 12.0 / 12.3 | 0.746 ± 0.015 | 3.0 | 64 | 76.0 ± 11.3 / 11.6 | 0.756 ± 0.016 | 2.6 |
| | 50 x 50 | 168 | 75.6 ± 9.7/ 10.0 | 0.738 ± 0.013 | 2.4 | 154 | 76.2 ± 8.5 / 8.9 | 0.747 ± 0.014 | 1.9 |
| | 25 x 25 | 678 | 76.8 ± 5.9 / 6.4 | 0.733 ± 0.009 | 2.0 | | | | |
| Cot2a (late fracture) | 200 x 200 | 12 | 16.6 ± 6.6 / 6.6 | 0.827 ± 0.0061 | 0.76 | 9 | 14.0 ± 6.1 / 6.1 | 0.825 ± 0.0057 | 0.44 |
| | 150 x 150 | 18 | 15.1 ± 3.7 / 3.7 | 0.825 ± 0.0041 | 1.2 | 18 | 15.1 ± 3.7 / 3.7 | 0.825 ± 0.0041 | 1.2 |
| | 100 x 100 | 31 | 14.1 ± 2.8 / 2.8 | 0.824 ± 0.0037 | 1.2 | 30 | 11.6 ± 1.8 / 1.8 | 0.824 ± 00.31 | 0.84 |
| | 75 x 75 | 46 | 14.1 ± 2.9 / 2.9 | 0.825 ± 0.0035 | 1.2 | 42 | 12.9 ± 2.6 / 2.6 | 0.823 ± 0.0034 | 0.92 |
| | 50 x 50 | 84 | 13.5 ± 2.3 / 2.3 | 0.823 ± 0.0036 | 1.3 | 79 | 13.5 ± 1.9 / 1.9 | 0.825 ± 0.0031 | 1.2 |
| | 25 x 25 | 240 | 12.7 ± 1.5 / 1.6 | 0.823 ± 0.0032 | 1.4 | | | | |
| DBT (anchored to 0.74) | 200 x 200 | 9 | 62.9 ± 2.0 / 2.9 | 0.74 | 3 | 8 | 63.5 ± 1.0 / 2.3 | 0.74 | 0.8 |
| | 150 x 150 | 13 | 63.0 ± 0.9 / 2.3 | 0.74 | 1 | 13 | 63.0 ± 0.9 / 2.3 | 0.74 | 1 |
| | 100 x 100 | 38 | 63.1 ± 0.9 / 2.3 | 0.74 | 1.7 | 36 | 63.0 ± 0.7 / 2.2 | 0.74 | 1.3 |
| | 75 x 75 | 50 | 63.2 ± 0.9 / 1.9 | 0.74 | 1.7 | 48 | 63.2 ± 0.7 / 2.2 | 0.74 | 1 |
| | 50 x 50 | 152 | 63.0 ± 0.6 / 2.2 | 0.74 | 1.4 | 148 | 63.0 ± 0.6 / 2.2 | 0.74 | 1.3 |
| | 25 x 25 | 608 | 62.5 ± 0.5 / 2.1 | 0.74 | 1.4 | | | | |
| Senz7 (self corrected)[a] | 200 x 200 | 6 | 152.6 ± 12.9 / 13.9 | 0.940 ± 0.250 | 0.80 | 6 | 152.6 ± 12.9 / 13.9 | 0.94 ± 0.25 | 0.80 |
| | 150 x 150 | 8 | 156.4 ± 9.2 / 10.6 | 1.030 ± 0.210 | 0.36 | 7 | 153.2 ± 8.1 / 9.6 | 0.98 ± 0.16 | 0.36 |
| | 100 x 100 | 24 | 153.4 ± 4.7 / 6.9 | 0.967 ± 0.093 | 0.87 | 22 | 146.3 ± 3.8 / 6.2 | 0.826 ± 0.051 | 0.53 |



| | | | | | | | | |
|---|---|---|---|---|---|---|---|---|
| | 75 x 75 | 32 | 147.4 ± 3.2 / 5.8 | 0.840 ± 0.047 | 1.0 | 30 | 145.8 ± 3.2 / 5.8 | 0.809 ± 0.040 | 1.1 |
| | 50 x 50 | 96 | 147.0 ± 2.1 / 5.3 | 0.821 ± 0.029 | 1.1 | 92 | 146.0 ± 2.0 / 5.2 | 0.811 ± 0.027 | 1.2 |
| | 25 x 25[b] | 384 | 147.3 ± 1.8 / 5.1 | 0.823 ± 0.027 | 1.6 | | | | |

**Table 3: Age and common Pb values obtained with the mobile grid method, for different virtual spot sizes, using the TW regression. The results presented are those obtained with the default grid position, and those corresponding to the best results in terms of precision and MSWD. [a]based on 50 μm x 50 μm virtual spots; [b]One ellipse removed**








**Figure 3: Tera-Wasserburg diagrams obtained for 5 samples (ARB20-2D, AUB-B6, BH14, DBT, Senz7 from top to bottom) for spot sizes of 200 μm x 200 μm, 100 μm x 100 μm, 50 μm x 50 μm and 25 μm x 25 μm (except sample AUG-B6: 150 μm x 150 μm, 75 μm x 75 μm, 50 μm x 50 μm and 25 μm x 25 μm). The diagrams correspond to the grid positions giving the best results in terms of precision and MSWD.**






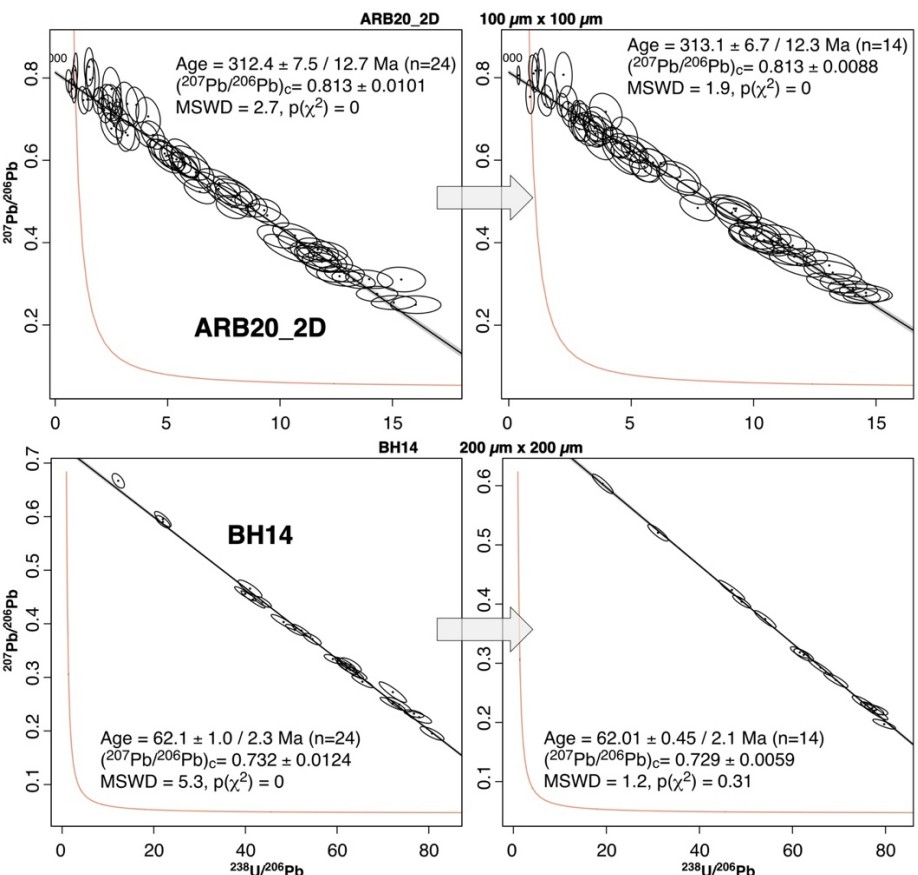

**Figure 4: Examples of Tera-Wasserburg diagrams obtained for ARB20-2D and BH14 with the default grid position (left), and those**
**corresponding to the best results in terms of precision and MSWD (right).**

**5.1.2 Case of low-U sample**

In the case of samples with low U (and Pb) contents (C6-265-D5), the ages calculated from isotopic image maps are clearly
biased. First, for 200 and 150 µm spots, the uncertainties on the ages obtained with the TW and concordia regressions are high
(higher than 15 Ma without propagation; Table 3), and the ages vary widely (from 0 Ma to > 200 Ma). For concordia
regressions, they correspond to errorchrons in nearly all grid positions. For lower spot sizes, the ages obtained are not identical
in the uncertainties, and vary according to the spot size. They are much lower in the case of TW regressions than in that of
concordia ones (Fig. 5A). The ages calculated by TW regression are centered around ~52-85 Ma (100 µm spots) and ~67-86
Ma (50 µm spots). Concordia regression results in ages centered around ~93-155 Ma (100 µm spots) and ~93-119 Ma (50 µm
spots) (Fig. 5C). These biased values are clearly related to the low number of counts in U and Pb, which induces a large
dispersion of isotope ratios (Fig. S2). They mostly result in high MSWD values. These inconsistencies are resolved by
averaging the number of mass sweeps on the time-resolved signal before data processing (equivalent to averaging the number



of pixels along each linear scan), as done by Drost et al. (2018) and Hoareau et al. (2021) (Fig. 5B). As shown in Fig. 5C, the mean TW and concordia ages evolve as a function of the number of averaged pixels, reaching identical values in their uncertainties from a value of 2, and a more restricted range from a value of 3. We note that the calculated average values vary slightly depending on the chosen spot size (~110-120 Ma for 100 µm and ~100-110 Ma for 50 µm). Despite their high uncertainties, these ages are consistent with the geological evidence of precipitation in the interval between ~120 and ~100 Ma (Pyrenean rifting; Motte et al., 2021).





Figure 5: A. Best age results obtained for C6-265-D5 sample, with virtual spot sizes of 100 μm x 100 μm and no mass sweep averaging (left: TW diagram; right: Wetherill diagram). B. Same as A but with four averaged mass sweeps. C. Evolution of the weighted mean age calculated from several grid positions and for several virtual spot sizes (50 μm x 50 μm and 100 μm x 100 μm), as a function of the number of averaged mass sweeps, for sample C6-265-D5. The vertical bars correspond to 95% standard deviation. The expected age is between 100 and 120 Ma (dashed lines).



## 5.2 Ages calculated with the 'Rectis' method

Using the Rectis method also yields very satisfactory results for high-U samples. In accordance with the theory for a sample of homogeneous age, the representation of all possible ellipses in a TW diagram defines a linear trend, with MSWD values close to or below 1 (Figs. 6B and S5). The ages obtained by orthogonal regression through the set of virtual spots are identical to the expected ones. After selecting the maximum number of non-adjacent 100 µm or 50 µm spots on an image map (starting from the 50% spots closest to the regression line), expected ages are also obtained with IsoplotR. For several samples (ARB20-2D, BH14, DBT, Senz7), besides the low MSWD values, the age uncertainties are better than those obtained by the mobile grid method (Fig. 6C; Table 4). For AUG_B6 and Cot02a samples, however, they are higher. In these cases, the virtual spots selected by Rectis may correspond to less spread ellipses in a TW diagram, and/or greater individual uncertainties. Regarding C6-265-D5, a similar behaviour to that observed with the grid method is obtained. The orthogonal regression gives an age of 72.6 Ma in the TW diagram, with a high MSWD (2.8) indicating the poor alignment of all ellipses (Fig. 6C). The corresponding age after selecting the virtual spots is 68.6 ± 9.8 Ma. With a Wetherill diagram, the ages are much higher (126.9 Ma for orthogonal regression and 126.1 ± 17.1 Ma after selection). Averaging the number of mass sweeps by 4 results in final ages of 120.3 ± 14.3 Ma and 112.3 ± 14.3 Ma (TW and Wetherill diagrams, respectively), values that are in better accordance with the expected age, while being associated with lower MSWD values (Fig. 6D). It should be noted that in this case the uncertainties are also lower than those obtained using the mobile grid method.





**Figure 6: A.** $^{238}$U/$^{206}$Pb map of ARB20-2D, and location of the maximum number of non-adjacent 100 μm x 100 μm spots as calculated with the Rectis method. **B.** Results obtained for ARB20-2D sample. Left: TW diagram built from all possible 100 μm x 100 μm virtual spot positions on the image (uncertainties not represented). Age, common Pb and MSWD values are calculated from *Scipy* ODR regression. Right: Age result obtained from the maximum number of non-adjacent spots using the Rectis method, starting from the 50% spots closest to the ODR regression line. **C.** Same as B but for C6-265-D5 sample. **D.** Same as C but with four averaged mass sweeps. See text for details.



| Sample name | Virtual spot size (μm) | Orthogonal regression (all possible virtual spots) | | | | Results from best virtual spot location | | | |
| --- | --- | --- | --- | --- | --- | --- | --- | --- | --- |
| | | Number of virtual spots | Age (Ma) | Pb0 | MSWD | Number of virtual spots | Age (without/with systematic uncertainty) (Ma) | Pb0 | MSWD |
| ARB20-2D | 100 x 100 | 12255 | 313.3 | 0.815 | 1.7 | 57 | 312.2 ± 5.4 / 11.6 | 0.814 ± 0.006 | 1.2 |
| AUG-B6 | 50 x 50 | 4417 | 43.5 | 0.853 | 1.4 | 61 | 44.1 ± 1.8 / 2.4 | 0.862 ± 0.016 | 0.71 |
| BH14 | 100 x 100 | 5627 | 62.2 | 0.730 | 0.8 | 47 | 62.2 ± 0.4 / 2.1 | 0.731 ± 0.004 | 0.84 |
| C6-265-D5 | 100 x 100 | 7455 | 72.6 | 0.752 | 2.8 | 33 | 68.6 ± 9.5 / 9.8 | 0.742 ± 0.020 | 1.0 |
| Cot2a (late fracture)[a] | 25 x 25 | 1817 | 12.9 | 0.822 | 1.4 | 112 | 13.9 ± 3.2 / 3.2 | 0.825 ± 0.003 | 0.28 |
| DBT (anchored to 0.74) | 100 x 100 | 2522 | 63.3 | 0.74 | 1.2 | 27 | 63.4 ± 0.8 / 2.3 | 0.74 | 0.30 |
| Senz7 (self corrected)[b] | 50 x 50 | 5887 | 147.2 | 0.831 | 0.7 | 81 | 147.3 ± 2.3 / 5.4 | 0.830 ± 0.033 | 0.22 |

**Table 4: Age and common Pb values obtained with the Rectis method, using the TW regression. The results presented are those**
**obtained by orthogonal regression (ODR) across all possible virtual spots, and after selecting the maximum number of non-adjacent spots on an image map. [a]Uncertainties greater than 20% filtered out; [b]based on 50 μm x 50 μm virtual spots.**



### 5.3 Calculation of ages from very small areas

### 5.3.1 Manual selection of cement phases (Cot02a)

The isotopic image mapping obtained on sample Cot02a suggests the presence of several generations of calcite in addition to
detrital quartz grains. Indeed, U and Th concentration maps highlight 2 distinct calcite generations carrying the highest U/Pb
ratios, which were manually selected using Iolite4 (Fig. 7A). The first phase consists of 2 grains adjacent to quartz, covering
a total area of approximately 0.064 mm$^2$ (equivalent to ~6 static spots of 100 μm in diameter), or ~2200 pixels. The average
U content is $1.5 \pm 0.03$ ppm. The second cement phase fills a late fracture of about 50 μm wide, cutting across the previous
phases as well as the quartz. The total selected area is approximately 0.092 mm$^2$ (i.e., ~9 static spots of 100 μm in diameter),
or ~3800 pixels. U contents are high, with an average value of $6.9 \pm 0.2$ ppm. For the first phase, the $^{238}$U/$^{206}$Pb ratios are very
low, resulting in very high uncertainties in the calculated ages (~25-40 Ma for ages centred around ~110-140 Ma, for virtual
spots of 50 μm) (Fig. 7B). As before, reducing the size of the virtual spots and therefore increasing their number helps decrease
the age uncertainty, which nevertheless remains above 15 Ma for 25 μm spots. Since virtual spots may only partially overlap
the selected areas, the number of spots may exceed what is expected based on the ratio between the selected cement surface
area and the virtual spot area. Thus, it is possible to obtain up to 135 spots or 270 spots of 25 μm x 25 μm or 12.5 μm x 25
μm, respectively, on the selected surface, but with individual uncertainties in the $^{238}$U/$^{206}$Pb and $^{207}$Pb/$^{206}$Pb ratios that can
sometimes exceed 50%. Filtering out uncertainties above 20% (as example) reduces the final number of spots but allows
obtaining close age values regardless of the position of the virtual grid (~90-110 $\pm$ 20 Ma) (Fig. 7B, 7C). Additionally, filtering
out spots containing too many pixels outside the selected zone (i.e., spots partially overflowing from this zone and thus with a
high number of unused pixels) results in close results, but with MSWD values closer to unity (Fig. 7C). Due to low $^{238}$U/$^{206}$Pb
values, such ages are roughly consistent with the stratigraphic age of the host carbonate (Coniacian-Santonian, i.e. 90-84 Ma).
However, a more precise age of 82 Ma $\pm$ 7.6 Ma was obtained from similar grains from another image made on the same
sample (not presented here), suggesting that in fact depositional carbonate or early calcite cement was remobilized by the shear
band formed during the Pyrenean orogeny. For the second cement phase, a similar methodology yields much more recent ages
(~12-14 Ma), in accordance with the petrographic evidence of late-stage fracturing (Fig. 7D). It is interesting to note that for
very small spot sizes (e.g., 12.5 μm x 25 μm), filtering out spots with uncertainties in the ratios above, for example, 10%
logically reduces the number of spots significantly (from 436 to 32), but provides age uncertainties around 2 Ma, which are
generally comparable to those obtained when all spots are retained (Fig. 7E). This allows testing various configurations (virtual
spot sizes, filtering or not, grid migration) to find the parameters that yield the most reliable ages possible.





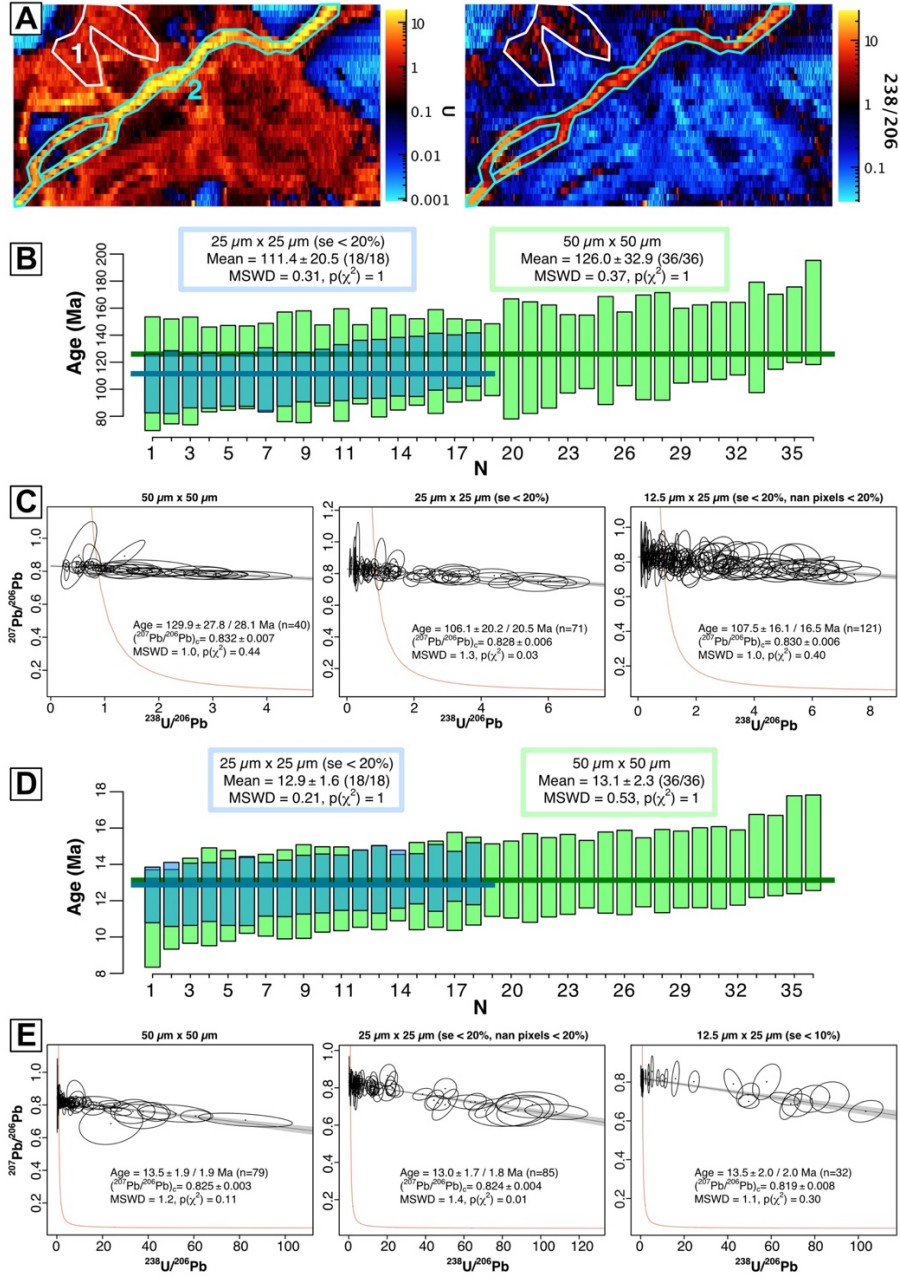

**Figure 7: A.** LA-ICP-MS maps of U concentrations (in ppm), and $^{238}$U/$^{206}$Pb ratios of sample Cot02a. The area highlighted in white ("1") corresponds to the first generation of calcite, while the area highlighted in blue ("2") corresponds to the second. **B.** Weighted average of ages obtained for the first generation, for different grid positions and virtual spot sizes. In blue, spots with uncertainties greater than 20% have been filtered out. **C.** Best age results obtained for different virtual spot sizes, for the first generation. For 25 μm spots, spots with more than 20% of pixels outside the selected area (see fig. 6A) are excluded, in addition to uncertainties greater than 20%. For 12.5 μm x 25 μm spots, the filtering is based on uncertainties greater than 10%. **D.** Same a B, but for the second generation. **E.** Same a C, but for the second generation. For 25 μm spots, spots with more than 20% of pixels outside the selected area and uncertainties greater than 20% are filtered out, while for 12.5 μm x 25 μm spots, spots with uncertainties greater than 10% are excluded.



### 5.3.2 Micro-images ('sub-image' method) and weighted average of ages

Here we use the example of samples ARB20-2D and BH14, which are highly suitable for dating due to their high U content and good spread of ratio values, to demonstrate the possibility of obtaining ages from isotopic mapping of extremely small surfaces, comparable to that of a single spot in conventional LA-ICP-MS approach, and of calculating weighted mean ages for a larger image. The isotopic map obtained on ARB20-2D covers an area of ~0.72 mm$^2$ (20488 pixels), which corresponds to 18, 72, 290 and 1160 virtual spots of 200 µm, 100 µm, 50 µm and 25 µm on each side, respectively. By choosing a 25 µm grid, the map can then be divided, for example, into 18 sub-images of 200 µm x 200 µm, each containing 64 virtual spots of 25 µm x 25 µm, or into 72 sub-images of 100 µm x 100 µm, each containing 16 virtual spots (Fig. 8A). In the case of 200 µm sub-images, the ages calculated for each sub-image are mostly comparable to the age obtained from the entire image using 25 µm spots (~305.7 ± 3.3 / 8.7 Ma) within the limits of uncertainties, with uncertainties that can be below 10%. The weighted mean age (306.6 ± 3.1 Ma without propagated uncertainties) is also identical to the expected age (Fig. 8B). The precision of the ages can be improved by choosing even smaller spot sizes (12.5 µm x 25 µm), corresponding to 128 virtual spots per 200 µm x 200 µm sub-image (Fig. 8C). In this case, 90% of the ages have uncertainties below 10%, with a comparable weighted mean (300.0 ± 3.1 Ma). Additionally, choosing 50 µm virtual spots provides greater accuracy in the calculated ages. In the case of 100 µm sub-images, the obtained ages have higher uncertainties, but the weighted mean is close (307.0 ± 3.4 Ma and 301.1 ± 3.2 Ma for 25 µm x 25 µm and 12.5 µm x 25 µm spots, respectively; Fig. 8B). However, it should be noted that in all cases, several ages from the sub-images differ from the expected age within the limits of uncertainties. By following the same procedure for sample BH14, the obtained ages for most sub-images are mostly identical to the expected age, with uncertainties generally better than 10% (Fig. 9). The weighted means are also perfectly comparable to the expected age. Again, choosing 50 µm spots in 200 µm sub-images yields the most reliable results (all ages are comparable to the expected age). In conclusion, these examples show that for favourable samples (U > 1 ppm, variable U/Pb), dating from 100 µm or 200 µm images is possible, with slightly lower precision and, to a lesser extent, accuracy compared to more conventional approaches. The calculation of weighted mean ages also allows controlling the quality of the age obtained from the isotopic maps.





**Figure 8: A. Scheme illustrating the procedure used to obtain ages from sub-images, based on ARB20-2D sample. A 100 μm x 100 μm (or 200 μm x 200 μm) image is extracted from the initial isotope map. It is itself discretised into small virtual spots (typically 25 μm x 25 μm). Two TW diagrams obtained from 2 different sub-images are shown as examples. B. Weighted averages of ages obtained from different sub-images for ARB20-2D, for 25 μm x 25 μm virtual spots. In blue, 200 μm sub-images and in green, 100 μm sub-images. Systematic uncertainties are not considered. C. TW diagrams of the best results obtained for different sub-image and virtual spot sizes. Note the good distribution of ellipses despite the small size of the sub-images, and the similarity of the calculated ages.**



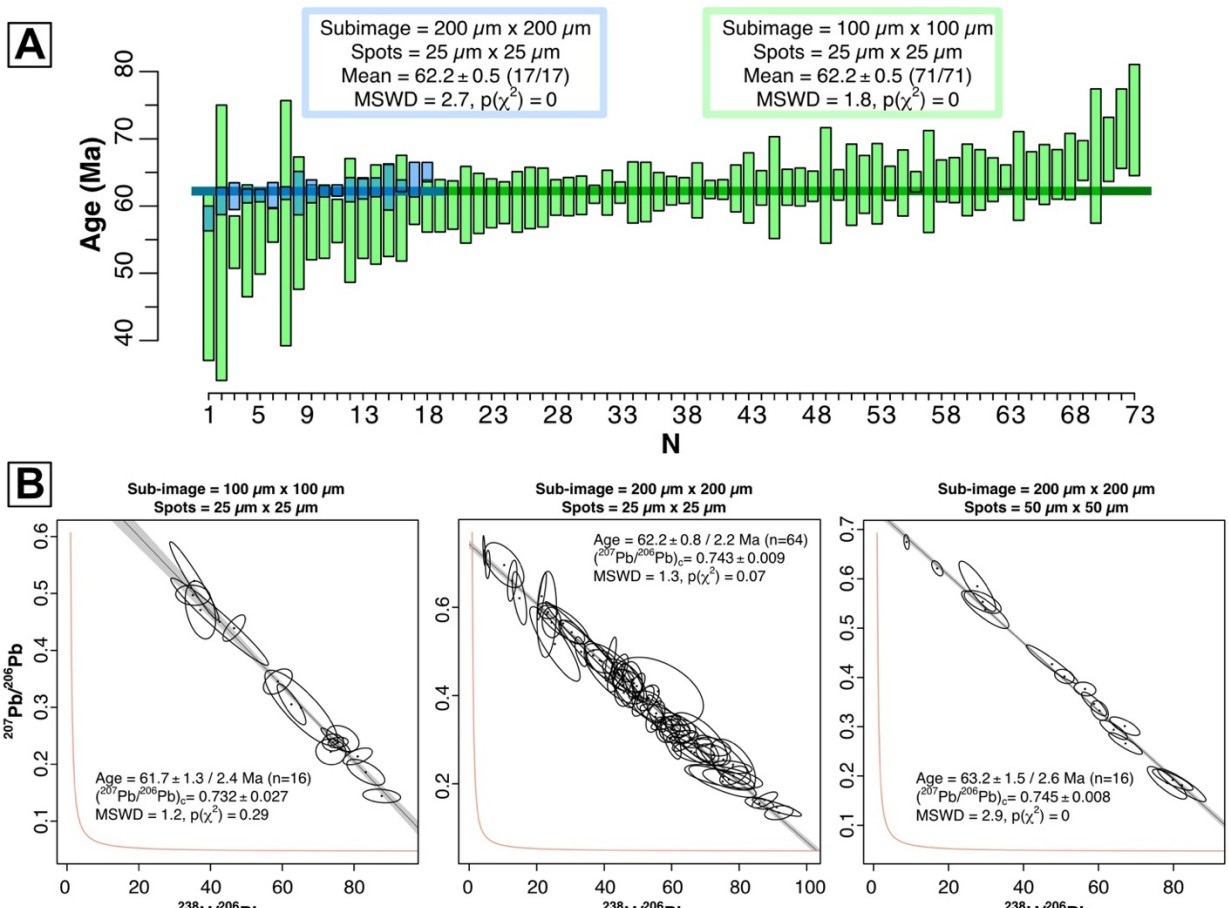

**Figure 9. A. Weighted averages of ages obtained from different sub-images for BH14, for 25 μm x 25 μm virtual spots. In blue, 200 μm sub-images and in green, 100 μm sub-images. Systematic uncertainties are not considered. B. TW diagrams of the best results obtained for different sub-image and virtual spot sizes. Note the good distribution of ellipses despite the small size of the sub-images, and the similarity of the calculated ages.**

## 6 Discussion

### 6.1 Interest and limitations of the method

The interest and inherent limitations of extracting ages from isotopic maps by LA-ICP-MS have already been addressed by several authors (Drost et al., 2018; Roberts et al., 2020; Hoareau et al., 2021; Davis and Rochín-Bañaga, 2021; Liu et al., 2023). One of the main advantages of isotopic maps, as detailed by Drost et al. (2018) and Roberts et al. (2020), is the ability to isolate pixels (either individually or as a range) based on their associated compositions or isotopic ratios, for example, to highlight multiple generations of cements or reject pixels with composition anomalies that may indicate detrital contamination before calculating one or more ages. Moreover, it is always possible to isolate ranges using co-localization approaches





(Hoareau et al., 2021) or simply by manual selection in suitable software (here, Iolite4; see also Ansberque et al., 2020; Chew et al., 2020). Various data processing methods for age calculations on carbonates have been proposed since the study by Drost et al. (2018). These include the pixel pooling method (Drost et al., 2018), robust regression on the values of the isotopic ratios of interest (Hoareau et al., 2021), as well as Bayesian regression (Davis and Rochín-Bañaga, 2021; Liu et al., 2023). As already discussed in the introduction, each approach has its advantages and disadvantages, and the present approach is no exception
to this rule.

On the one hand, the results presented here show that for samples traditionally favorable for U-Pb dating of carbonates by LA-ICP-MS (U > 100 ppb, good spread of U/Pb ratios of pixels), the virtual spot approach has several interesting advantages, especially when used on data obtained with a high repetition rate laser that allows high spatial resolutions (here, pixels of 25 µm x 1.3 µm), coupled with a high-sensitivity spectrometer (here, an SF-ICP-MS). In fact, it is possible, from isotopic maps
typically obtained in less than 1 hour, to generate hundreds of spots covering the selected carbonate phases. The ability to adjust the spot size allows finding the most suitable combination in terms of statistics on the considered cement (low age uncertainty, MSWD close to 1), while avoiding possible accuracy biases through alternative approaches. The example of the Cot6a sample shows that robust ages are obtained on a mineralized fracture less than 50 µm thick, which would have been challenging to date with conventional static spot approaches. Here, the ability to generate more than 100 spots smaller than 50
µm is a notable advantage. Generally, spot sizes of 50 µm x 50 µm yield very satisfactory results. Another development presented here is the use of a mobile grid for virtual spots, which allows the user to calculate several tens of ages from the same image to evaluate their relevance, for example, through the visualization of their weighted means. In the case of reliable samples, it is expected that the position of the spots does not influence the calculated ages. It is then possible to choose the final age with the best statistical parameters. The Rectis method offers an alternative approach to age calculation that can
results in even better statistical results. Finally, the approach allows for age calculation from very small image dimensions (down to 100 µm x 100 µm), although with the limitations presented with the examples of samples ARB20-2D and BH14. For larger images, dividing them into sub-images allows the calculation of weighted mean ages, providing an additional way to evaluate the relevance of a cement age.

On the other hand, the virtual spot approach is not flawless. First, the results obtained on sample C6-265-D5 show that a bias
towards a too young age is possible when U and/or Pb concentrations are too low. It is then necessary to average the pixels (as done by Drost et al., 2018) and compare the ages obtained on TW and concordia diagrams to obtain a satisfactory result, which can be laborious. Even with such pixel averaging, tests carried out on the ASH15 standard (Nuriel et al., 2021) failed to obtain a reliable age due to the very low Pb content of the sample (< 7 ppb), although this standard is used by several teams equipped with an identical mass spectrometer (e.g., Montano et al., 2021; Guillong et al., 2020). It therefore appears that this problem
of bias requires further work on the conditions under which LA-ICP-MS images are acquired. Second, each virtual spot is obtained from adjacent pixels, rather than from the progressive ablation of the same surface as in the case of static spots. Given the variation of isotopic ratio values at the microscopic scale in carbonates, it is expected that the uncertainty of the average obtained for each virtual spot will be larger. This limitation can be counterbalanced by using more virtual spots, as shown by



the results obtained, for example, on samples BH14, DBT, AUG-B6, and ARB20-2D. Another counterintuitive effect of using
virtual spots is the occurrence of high MSWD values for samples particularly favourable for dating, due to high error
correlations. Sample BH14 is representative of this effect. For each virtual spot (100 μm x 100 μm), the variations in the
$^{238}$U/$^{206}$Pb and $^{207}$Pb/$^{206}$Pb ratios are significant, resulting in high error correlations. This characteristic is likely to highlight
minor heterogeneities in the sample. These indeed seem to exist in sample BH14 as shown by the MSWD value of 4.7 also
obtained by Hoareau et al. (2021) with static spots of same size, higher than the value of 1.6 of Beaudoin et al. (2018). Note
that this error correlation effect can also be considered as an additional means of better characterizing the sample in question,
in favour of the approach presented here.

**6.2 Further developments**

Further developments can be envisaged for the virtual spot method, to further enhance its appeal. A first major
development will be to propose a routine that automatically selects virtual spots for maximum spread of isotope ratio values,
for the number of virtual spots chosen by the user. This should lead to improved accuracy and precision of calculated ages.
Other developments could include the automatic definition of virtual spots of variable size and shape, depending on the
precision of the resulting isotope ratios, with a view to improving age calculations. Finally, in a more fundamental sense, the
use of an average and its uncertainty implies that the distribution of pixel isotopic ratio values follows a normal distribution
for each spot, which is probably valid only for the largest spots due to their higher number of pixels. The similarity and
accuracy of the ages obtained here for different virtual spot sizes shows that skewed pixel values distribution does not introduce
measurable bias into the results. However, additional tests must focus on using alternative statistics such as for example the
median and its uncertainty.

**7 Conclusion**

The U-Pb carbonate dating approach from isotopic maps presented here takes benefit from the use of a high sensibility ICP-
SF-MS coupled with a high-repetition-rate ablation laser. The high ablation rate (> 100 Hz) enables high spatial resolution
maps to be obtained, which are then simply separated into a grid of virtual spots for the calculation of ages. Tests carried out
on samples of known age show that the ages obtained correspond to reliable ages unbiased by the processing method. For
samples with low U concentrations and noisy U/Pb and Pb/Pb ratios, comparison of the ages obtained from the TW and
concordia diagrams, together with smoothing of the number of pixels in the isotopic images, corrects the bias towards too-
young ages obtained from the unsmoothed maps. In addition to the advantages inherent to the use of isotopic maps described
elsewhere (such as the possibility of filtering pixels or manually selecting regions of interest), the ability to move virtual spot
grids implies that several ages can be obtained for the same study area, helping to assess their homogeneity. Finally, in the
case of U-rich samples, we show that one can obtain reliable ages from very small images (< 0.04 mm$^2$), paving the way for
dating samples traditionally inaccessible to geochronology, such as micro-veins or micro-fossils. Although only tested on



carbonates, there are no a priori limitations to the use of the virtual spot approach on other minerals traditionally used in U-Pb geochronology.

**Appendix A**

**Table A1: Analytical conditions**

| Laboratory & Sample Preparation | |
|---|---|
| Laboratory name | Institut des sciences analytiques et de physico-chimie pour l'environnement et les matériaux (IPREM), UPPA, Pau (France) |
| Sample type/mineral | Calcite / dolomite |
| Sample preparation | In situ in polished blocks or thin sections (30 µm) |
| Imaging | Yes |
| Laser ablation system | |
| Make, Model & type | Lambda 3, Nexeya (France) |
| Ablation cell | Home-made (home-designed) two volumes ablation cell. The large cell has a rectangular shape and a volume of 11.25 cm$^3$ (75 x 25 x 6 mm size) while the small one, placed above the sample is of 10 mm diameter. |
| Laser wavelength (nm) | 257 nm |
| Pulse duration (fs) | 360 fs |
| Fluence (J.cm$^{-2}$) | 5-8 J.cm$^{-2}$ |
| Repetition rate (Hz) | 100 or 500 Hz |
| Gas blank (s) | 15 s per image (1 line) |
| Ablation duration (s) | 38.3 to 145 s per line |
| Washout and/or travel time in between analyses (s) | Wash out time: ~1000 ms (Ar, October 2018) or ~500 ms (He, all other sessions). 15 vs or 25 s of break between lines to allow data processing. |
| Spot diameter (µm) | 15 µm |
| Sampling mode / pattern | Ablation lines (25 µm-height) made by combining laser beam movement across the surface (5 mm/s) and stage movement (25 µm/s). 25 µm between lines. |
| Cell Carrier gas (L/min) | He = 0.600 L/min |





| ICP-MS Instrument | |
|---|---|
| Make, Model & type | ICPMS Thermo Fisher ElementXR HR Jet Interface |
| RF power (W) | 1000 - 1100W |
| Cooling gas flow rate | 16 L min$^{-1}$ |
| Auxiliary gas flow rate | 1 L min$^{-1}$ |
| Nebuliser gas flow rate | 0.5 L min$^{-1}$ |
| Masses measured | 206, 207, 208, 232, 238 |
| Samples per peak | 30 |
| Mass window | 10 % |
| Sample time | 3 ms |
| Settling time | 1 ms |
| Mass sweep | 57 ms (most samples) |
| Averaged mass sweep | No |
| Resolution | 300 |
| Sensitivity | Percentage of ions detected with regard to atoms ablated is ~0.04% for U, as calculated with NIST 614 |
| Data Processing | |
| Calibration strategy | Calibration by standard bracketing; NIST614 for Pb-Pb and WC-1 calcite for Pb-U |
| Reference Material info | Primary: NIST612 (before 2020) and NIST614 - Woodhead and Hergt (2001) <br> WC-: 254.4 ± 6.4 Ma (2s) - Roberts et al., 2017 <br> Secondary: Duff Brown 64.04 ± 0.67 Ma (2s) - Hill et al., 2016 <br> AUG-B6 ~42.5 Ma – Pagel et al., 2018; Blaise et al., 2023 |
| Data processing package used / Correction for LIEF | Element XR acquisition software, data processing with Iolite 4 and in-house Python/R code. Age determination through virtual spot discretization. |
| Common-Pb correction, | No common Pb correction. Ages in the figures are quoted at 95% absolute uncertainties and include systematic uncertainties (WC1 2.7%, decay constants 0.1%, long-term uncertainty 2%), propagation is by quadratic addition. |



| composition and uncertainty | |
|---|---|
| Quality control / Validation | 3 analyses of Duff Brown (anchored to common Pb value of 0.74) gave ages of 62.6 ± 1.8 Ma, 60.7 ± 1.7 Ma, 65.9 ± 1.8 Ma. One analysis of AUG-B6 gave 42.2 ± 2.5 Ma. |

**Data availability.**

The Supplementary material (methodology for LA-ICP-MS spot analyses, additional plots, pixel values of the isotopic images and Python / R codes used for the data treatment) are publicly available in a Zenodo repository at https://doi.org/10.5281/zenodo.12820356.

**Supplement.**

The supplement related to this article is available online at https://doi.org/10.5281/zenodo.12820356.

**Author contributions.**

**Guilhem Hoareau:** Conceptualization, Data curation, Formal analysis, Investigation, Methodology, Project administration, Software, Supervision, Validation, Visualization, Writing - original draft, Writing - review & editing. **Fanny Claverie:** Investigation, Methodology, Resources, Project administration, Validation, Visualization, Writing - original draft, Writing -
review & editing. **Christophe Pecheyran:** Conceptualization, Investigation, Methodology, Project administration, Supervision, Validation, Visualization, Writing - review & editing. **Gaëlle Barbotin:** Investigation, Methodology, Resources, Validation, Visualization. **Mickael Perk:** Data curation, Formal analysis, Methodology, Resources, Software, Writing - original draft, Writing - review & editing. **Nicolas E. Beaudoin:** Investigation, Resources, Validation, Visualization, Writing - original draft, Writing - review & editing. **Brice Lacroix**: Investigation, Resources, Validation, Writing - original draft,
Writing - review & editing. **E. Troy Rasbury**: Formal analysis, Investigation, Resources, Validation, Visualization, Writing - original draft, Writing - review & editing

**Competing interests.**

The authors declare that they have no conflict of interest.



**Review statement.**


**Acknowledgements**

The idea of a mobile grid came out after a discussion with Julien Mercadier (CNRS, Nancy) during the Spectratom 2022 congress in Pau. Pr. Joseph Mitchell (Stony Brooks University, New-York) is warmly thanked for fruitful discussion regarding the non-overlapping rectangle problem.

**Financial support.**

The samples were analyzed by LA-ICP-MS or ID-MS-ICP-MS as part of several projects not directly related to the present work, which was not supported by any funding.

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
