# Peer review of "The virtual spot approach: a simple method for image U-Pb carbonate geochronology by high-repetition rate LA-ICP-MS."

_EGUsphere, 2024_

## Author Comment (AC1)

We thank the reviewer for their very detailed comments, which prompted us to reflect deeply on several of the points raised. We hope that our responses are sufficiently precise and well-supported. The original comments are in bold font, and the responses in normal font.

**General Comments:**

**This manuscript presents an approach to reduction of large quantities of spatially characterized isotopic data obtained by raster scans of samples using LA-ICPMS. Image reduction programs already exist. The main novel feature of this one seems to be the ability to average data from virtual spots in order to achieve an optimal balance between precision and spatial resolution. As such, it is of potential use to a large number of geochemists and geochronologists. I found it to be very interesting, although I have no experience in using a femtosecond laser and have never tried elemental imaging.**

We thank you for this overall positive comment.

**Specific Comments:**

**The manuscript could use a more extensive and clearer discussion of limitations and how these might be improved.**

The main limitation of our approach concerns samples with low U and/or Pb concentrations (low number of counts), for which biased ages may be obtained depending on the regression method used (TW or Wetherill). These points are discussed in more detail below. We will place greater emphasis on this issue and propose a workflow for applying the approach, following a suggestion from Barbara Kunz and Igor Figueiredo (reviewers 2).

**The authors use a single collector mass spectrometer so masses are measured at different times. The lateral movement between mass scans is about 1.4 mic while the pulse separation is about 0.05 mic. However, spatial resolution of data should be mainly limited by the ca. 0.6 sec transfer function (washout time) of ablated material into the plasma (in contrast to line 465), which covers a range of about 15 mic of beam movement along a scan. The observed distribution of isotopic ratios in the direction of a scan should be the actual distribution convoluted with the transfer function of the instrument. This would cause data variations along a scan at a scale lower than about 15 mic to be smeared together by the response time of the instrument, which is much broader than the distance covered by a mass sweep (1.4 mic).**

We agree that from an LA-ICP-MS imaging quality perspective, not accounting for the longer washout time when choosing the mass sweep duration (whether averaged to a higher value or not) can result in image smearing. However, Figure 2 shows that this smearing remains limited with a washout time of approximately 500 ms. Visual image resolution is not the primary objective in this case, although, of course, the highest possible resolution is desirable. The guiding principle here is to retain all measurements (each mass scan), even if signal mixing occurs, because the goal is to generate virtual spots, by analogy with static spot ablations. Signal mixing is common in static ablation, where a more homogeneous signal is sought to improve statistical reliability. In such cases, each ablation takes a different volume of the sample, but the overall signal is mixed/homogenized. A higher sampling time improves the statistical uncertainty on the mean, which decreases with the square root of N. In the case of the images produced here, the signal will not be as homogeneous as in static ablation, since the system's washout time remains relatively short (500 ms). This value represents a compromise between limiting image smearing along a line and obtaining a sufficiently reliable uncertainty on the mean. It should be noted that averaging the mass scans can have a beneficial effect on age accuracy for low-concentration samples, as illustrated by the example of C6-265-D5. Conversely, the example of BH14—based on both the results from Hoareau et al. (2021), who used an averaged mass scan of 540 ms, and the present study— shows that results are more satisfactory without averaging the mass scans. Therefore, for each sample, it is reasonable to test multiple configurations (with and without averaging) to select the most accurate and precise one.

**Therefore, defining the mass sweep as a pixel seems a bit misleading. Pixels, in the sense of a fundamental area scale at which independent values can be measured, are really more like the size of the spots but they vary along a continuum so it might be better to avoid the term pixel altogether.**

We initially used the term *pixel* in its general definition, which goes beyond isotopic imaging, namely 'the smallest addressable element in a raster image' (Wikipedia). This definition is independent of any potential signal correlation between adjacent pixels and is easily understandable, even for non-specialist readers. As an example, in the case of digital images, increasing the resolution from 100 px × 100 px to 200 px × 200 px multiplies the number of pixels by four based on the original image, with the new pixels necessarily being correlated (in terms of colour). Nevertheless, they are still referred to as pixels.

We prefer to retain the term *pixel* as a clear and accessible term, while adding a clarification in the Methods section of the manuscript to address the issues you raise from the perspective of LA-ICP-MS imaging.

**The relatively slow instrument response time is largely the result of using a nebulizer chamber where carrier He and makeup Ar are mixed before injection into the plasma. Response time can be significantly lowered by the use of a**

**modified, commercially available, quartz injection tube where mixing between the He carrier gas and Ar occurs within the plasma. Further improvement would require the use of a multi-collector or time of flight mass spectrometer in which signals for different masses are effectively measured simultaneously.**

We fully agree with these remarks, which go beyond the scope of the present study. The instrumentation used is not necessarily the most efficient for LA-ICP-MS imaging, and we hope to benefit from more advanced setups in the future.

**The scale of observable compositional variations across (rather than along) scans will be limited by the beam size (25 mic).**

Yes. To be more precise, the beam size is 15 µm but galvanometric scanners allow to rapidly move the beam to generate a 25 µm-width line, but the result is the same.

**It should also be noted that although the scan line is claimed to have a depth of 40 mic at 500 Hz, the troughs should have a V-shaped profile for non-overlapping lines so the average depth would be 20 mic. One should therefore be mapping triangular segments beneath the surface. This should of course affect ablation bias so standards need to be analyzed the same way.**

We fully agree. The troughs broadly have a V-shape, with the highest depth reaching about 40 µm as measured with a digital microscope. This maximum value is not an average. It is this maximum penetration depth that is critical for us when analysing thin samples, which must not be traversed. In any case, we always use strictly similar parameters to analyse the standards.

**A more fundamental problem is the fact that regression of relatively imprecise ratios produces ages that can be significantly different using the Wetherhill plot and the more commonly used Tera-Wasserburg (T-W) plot.**

We agree. This is clearly shown by the example of sample C6-265-D5.

**I assume the reason is that one measures counts on the masses but one plots and calculates with ratios of these counts. Errors in the numerator mass count will propagate linearly whereas those in the denominator mass count will only propagate linearly to first order so second order effects become important when the relative error is large. This was the main reason that I developed the approach of regressing signals in a 3D space where the Wetherhill solution generally agrees most closely with the 3D solution for poorly radiogenic data sets and both disagree for highly radiogenic data sets (see Davis and Rochan-Banaga, 2021, Fig. 5).**

Your comment is very interesting; indeed, we did not address this specific point, which you discuss in detail in Davis and Rochin-Banaga (2021). Clearly, this was not

an aspect we intended to cover here, as we are not able to provide further insights beyond those you have already offered. We tested sample C6-265-D5 using the Brama2.0 software developed by Liu et al. (2023), which is a Python-based equivalent of UtilChron. We were also unable to obtain a satisfactory age. The initial regression in Wetherill space yields an age of around 115 Ma, whereas the Bayesian solution gives an age of approximately 63 Ma. This discrepancy is not resolved by averaging the mass sweeps in groups of 8, for instance (as done in Hoareau et al. 2021).

**The reason can be visualized by considering that non-linear variations in the denominator isotope in T-W plots ($^{206}$Pb) will bias data along a diagonal, which usually is at a high angle to the mixing line (isochron), whereas in the Wetherhill plot the denominator isotopes are from U, which is usually the largest peak (lowest error) and the common Pb end member is at infinity so non-linear variations will tend to be sub-parallel to the mixing line unless the datum is very radiogenic. The manuscript demonstrates this clearly but does not offer any discussion on how to effectively deal with it. One way would be to do regressions in 3D but there may be ways of correcting for it in 2D.**

In the manuscript, we propose resolving these age inconsistencies between TW and Wetherill regressions by averaging the number of mass scans (smoothing), even though this results in larger individual uncertainties for the virtual spots. For easily datable samples (e.g., ARB20-2D or BH14), TW and Wetherill ages agree within uncertainties, and smoothing is unnecessary: it does not change age values but increases their uncertainty. For low-concentration samples (e.g., C6-265-D5), smoothing helps eliminate erratic isotopic ratios caused by too many zero or negative counts for $^{207}$Pb and $^{206}$Pb, which usually result in overly high $^{238}$U/$^{206}$Pb and $^{207}$Pb/$^{206}$Pb ratios in TW diagrams. These ratios bias the regression towards a shallower slope and thus a younger age. We agree that this effect should be less pronounced in Wetherill regressions. However, the C6-265-D5 example shows that averaging the mass scans alters both the TW and Wetherill ages, which then converge towards a common value. We consider this convergent value more reliable, although it should ideally be tested on samples of known age.

To further expand on this very interesting issue, we present here a point we discussed in 2024 with Pieter Vermeesch, concerning both the age differences between TW and Wetherill regressions and the discrepancies in results between Isoplot and IsoplotR. Taking sample C6-265-D5 as an example, without averaging the mass scans and using virtual spots of 100 µm × 100 µm across the entire image (n = 42), the age obtained with Isoplot from $^{206}$Pb/$^{238}$U and $^{207}$Pb/$^{235}$U ratios in Wetherill space is 140 ± 25 Ma, identical to the age calculated using IsoplotR. In contrast, the age obtained from $^{238}$U/$^{206}$Pb and $^{207}$Pb/$^{206}$Pb ratios in TW space using Isoplot is 84 ± 16 Ma, which is higher than the age proposed by IsoplotR (76 ± 17 Ma). This discrepancy stems from differences in the regression methods used by the two software packages:

1. In Isoplot, regressions—whether in TW or Wetherill space—are performed using the York algorithm. The software can compute four different ages for a single sample depending on the chosen ratios and coordinate space.
2. In contrast, IsoplotR always performs regressions in Wetherill space, regardless of the input ratios (whether $^{207}$Pb/$^{235}$U and $^{206}$Pb/$^{238}$U, or $^{238}$U/$^{206}$Pb and $^{207}$Pb/$^{206}$Pb). The software, which uses the York and/or Ludwig (1998) algorithm (both yielding identical results), thus calculates only two ages. The results are then simply displayed in either the TW or Wetherill plot, depending on the user's choice.

Therefore, if one uses $^{238}$U/$^{206}$Pb and $^{207}$Pb/$^{206}$Pb ratios to perform a Wetherill regression in Isoplot, the resulting value will be identical to the one obtained in TW coordinates using IsoplotR, i.e., 76 ± 17 Ma. For easily datable samples such as BH14, both software tools yield identical or nearly identical results regardless of the chosen method. The discrepancies observed in bad samples such as C6-265-D5, as noted by Pieter Vermeesch, illustrate that "*both age estimates are likely biased and wrong*". In our view, comparing TW and Wetherill age estimates should become a systematic prerequisite to assess the robustness of U-Pb geochronology results in general.

**The best application that I can think of for this method would be WC1. This is an excellent standard because of its relatively old age and high U concentration but shows evidence of not being homogeneous in age based on the high ID-TIMS age error (2.7%) and the work of Guilong et al (2020, https://doi.org/10.5194/gchron-2019-20). If it were possible to isolate the predominate phase, this would be a much more useful standard.**

This is an interesting comment, although we believe that our approach may also prove useful for other types of samples.

**Technical corrections:**

**If one choses not to indent paragraphs there should be a space left between them, as well as between references.**

OK. We will check this in the new version.

**In some places I found the phasing unclear or ambiguous. Suggestions for improvement are made on an annotated copy.**

Thank you for the time spent for the phrasing. We will try to follow all suggestions made.

**The authors refer to Wetherhill Concordia plots as 'concordia' and the inverse (but more commonly used in this application) Tera-Wasserburg concordia plot**

**as T-W. This is confusing because both are concordia plots. It would be better to refer to T-W and W plots.**

We agree that it is unclear as presented. We will make the appropriate changes.

**Line 62:**

**Presumably the authors mean 207Pb/235U, but why would one want to use this ratio, rather than the more precise 206Pb/238U ratio as a criterion for sorting? They both encode the same information (age and radiogenicity).**

We suggest to directly contact Kirsten Drost to discuss this, as it is not our proper work. The problem with $^{206}Pb/^{238}U$ sorting is that it tends to overweight outliers with high $^{238}U/^{206}Pb$ ratios in the TW space, resulting in results biased towards too young ages.

**Line 73:**

**The 3D regression also allows for editing outliers. This aspect is a separate problem from the best regression approach as discussed above.**

We agree. However, we wanted to emphasize that "bad quality samples" may also give biased results with the Bayesian approach: the prior estimate is obtained with a York regression on individual pixel values to which uncertainties are added from Poisson statistics. If the Isoplot estimate is biased for the reasons exposed above, the Bayesian result might be biased too, as we understand from our experience with this approach.

**Lines 125, 129, Table 1:**

**A better use of English would be to refer to line width (rather than height) as the diameter of the laser beam and line length (rather than width)  as the scanning distance. Otherwise it can be confusing to the reader.**

OK. We will change the terms in the next version of the manuscript.

---

## Author Comment (AC2)

We thank the reviewers for their very detailed comments, which prompted us to reflect deeply on several of the points raised. We hope that our responses are sufficiently precise and well-supported. The original comments are in bold font, and the responses in normal font.

**General Comments:**

**The manuscript presents a novel approach to obtaining U-Pb ages of carbonates using isotopic maps and guided by statistical approaches to obtain the best age in what can be considered virtual spots. This is an interesting approach, and we can see the benefits the approach can have. The authors present a large amount of data from a number of samples (some of which have been previously dated for comparison).**

**Specific Comments:**

- **In the introduction, the manuscript heavily relies on references and comparison with other studies, which when one isn't familiar with all of them makes reading and following the arguments a bit difficult. Particularly as the reader doesn't know the details of the approach this paper takes at that point in the text. Therefore, we wonder if some of this might be better suited for the discussion instead of the introduction. We would welcome it if the abstract and the introduction would focus a bit more on the rationale for using this approach. Why is this new method needed, what is the overall problem, that justifies using the approach the authors present? The text says that the ages are similar to the traditional approach but that the uncertainties can be worse. So, the authors should be clearer what the advantage of this method over the other methods is.**

In the introduction, we will aim to clarify the relevance of using isotopic imaging for geochronology, as well as the need to develop approaches that yield the most reliable ages possible. However, we do not intend to engage in a comparison of 'which data processing model is the best.' In this contribution, we present an alternative approach to those already published. We do not wish to portray our approach as a breakthrough that renders previous work obsolete. On the contrary, our method was developed through reflections inspired by existing approaches. As stated in the introduction and the discussion, each method has its own strengths and weaknesses, and ours is no exception. Through this contribution, above all we aim to share with the community the progress of our ongoing work on LA-ICP-MS map processing for U-Pb geochronology.

**After stating here in the methods that some samples have been treated differently none of this is mentioned afterwards in the results of discussion. Even if it doesn't make a difference. It would be important to mention that explicitly. If it does make a difference could some of the results be influenced by this and if yes how?**

We suppose you give details of the different treatments in the following lines.

- **Line 128: Why was the laser frequency changed from 100 to 500 Hz, please explain this change and the advantage of using one over the other. Additionally, can you detail if and what change this has on the results?**

In this contribution, we draw on analytical results obtained over the course of several years and across numerous research projects. By default, we work at 500 Hz on thick samples (mounts) to maximize signal intensity. With this setup, we typically obtain 100k–200k cps for $^{238}$U on the NIST SRM 614. The work performed at 100 Hz (ARB20-2D) was conducted on a thin section. The only reason for this was the shallower ablation depth, which prevents drilling through the section. It is of course associated with lower number of counts (70–100k cps). This change in repetition rate has no impact on how the data are processed using our method.

- **Line 137: For sample BH14 only Ar was used as a carrier gas. It has been shown by Eggins et al., 1998 that Ar and He have quite different transport qualities. How did the use of only Ar as a carrier gas influence the results?**

We cannot answer. BH14 was one of the very first samples analysed using imaging in our laboratory. We initially tested Ar to assess the feasibility of carbonate imaging, knowing that the washout time would be longer. Later, switching to He was motivated by its faster washout time, which provided greater responsiveness, although it also led to slightly more signal variability (based on standards WC1 and NIST), without any significant impact on the quality of the results in terms of precision and accuracy. We did not repeat tests using Ar after the analyses of BH14.

- **Line 149: Why was NIST SRM 614 used after 2020 and 612 before 2020? Are there any differences in the results or uncertainties?**

Initially, we worked with NIST SRM 612 for two main reasons: (i) it was already used to tune the spectrometer at the beginning of each session, making it a practical choice. Our custom-built ablation cell does not allow for multiple standards alongside the unknown sample, so keeping NIST SRM 612 saved time; and (ii) tests using NIST SRM 614 revealed slight inhomogeneities in $^{206}$Pb/$^{238}$U ratios (as has been reported elsewhere). However, since the concentrations in NIST SRM 614 are closer to those found in natural calcite, its use for calibration appears more appropriate

and justifies the switch. Objectively, we did not observe any significant differences depending on the standard used. It is also worth noting that some research groups use NIST SRM 612 for Pb/Pb ratio correction (e.g., Parrish et al., 2018).

- **Use of rainbow colour maps. Jet or rainbow colour maps have been shown to be not a good choice. Both in terms of accessibility (colour blind and other sight impairments) the rainbow scheme is also misleading normal normal-sighted people due to a higher sensitivity of the human eye for certain colours which leads to a visual bias. Therefore, it should not be used:**

  **https://blogs.egu.eu/divisions/gd/2017/08/23/the-rainbow-colour-map/**

  **Have a look here for alternative suggestions https://www.fabiocrameri.ch/colourmaps/**

Thank you for your suggestion. We will revise the color schemes in the figures to ensure they are accessible to visually impaired readers.

- **The comparison of the ages already determined by other studies (AUG-B6, BH14, DBT, Senz7) and this manuscript isn't very clear. In section 5.1.1. it is mentioned, but in none of the tables or figures (despite a reference to it) is it shown clearly.**

The reference ages are provided in Table 2 along with the appropriate citations, as well as in Section 2. In Section 5.1.1, Table 2 is indeed not referenced, which we will correct.

- **A lot of the ages are quoted without external error propagation. How much would external uncertainty add? I assume a good amount of external uncertainty comes from the standards? What other sources are there and what is the justification to ignore them? When comparing ages from previous studies with this one external error propagation would be needed to understand the full extent of how they compare to each other.**

We suppose that what you call 'external error' is that coming from the standards (referred to as "External 2σ err req'd (each pt)" in Isoplot). In our current procedure this additional uncertainty is directly calculated by Iolite4 from NIST SRM 614, based on the paper of Paton et al 2010 (reference will be added to the Methods section for clarity). Note that since we follow closely the recommendations of Horstwood et al. (2016), such excess variance is added directly onto the data points (ellipses) and not onto the final age. Additional systematic uncertainty includes a long-term variance (2%) that should allow confident comparison with ages obtained by other studies.

- **In section 5.2 the choice of data processing approach shows large (up to 50 Ma) variations in age (shown in figure 6 as well). How should a user choose which one to use? In the text, the authors mention that they chose the one closer to the expected age. But what if one doesn't know the expected age? Wouldn't this also mean that people might reject the 'true' age because they didn't expect it?**

Thank you for your pertinent observation. We fully agree that obtaining reliable ages from low-concentration samples—such as C6-265-D5—remains one of the main limitations of the approach presented in this study. It is indeed essential to verify whether a given sample is suitable for accurate U-Pb dating using this method. We address this issue in detail in our response to Don Davis, but we will also clarify it further in the revised manuscript. Specifically, we propose including a more explicit discussion outlining the importance of systematically comparing ages obtained from both Tera-Wasserburg and Wetherill regressions, in addition to evaluating conventional statistical parameters such as the MSWD. If the TW and Wetherill concordia ages do not overlap within their respective uncertainties, this should be taken as a strong indicator of potential bias and sample unreliability. Our tests suggest that, in such cases, averaging (smoothing) the mass scans can help reduce the impact of erratic isotope ratios, leading to better agreement between TW and Wetherill ages. However, this comes at the cost of reduced precision due to the lower number of pixels per virtual spot. Whether this trade-off improves accuracy remains an open question. The example of C6-265-D5 is promising, as the age obtained after smoothing aligns well with independent geological constraints. That said, we acknowledge that further work is needed to assess the robustness of this approach across a broader range of low-concentration samples. We will add this discussion to the revised manuscript, both to clarify the limitations and to outline potential future improvements.

- **In section 5.3.1 line 348 the authors refer to a map of sample Cot02a that has more precise ages, but then say it isn't presented here? Why is that? Why not at least provide this in the supplement?**

We did not include the second map (produced during the same analytical session) in the main manuscript because our objective here was to focus on the potential of isotopic mapping to distinguish and date multiple generations of calcite—specifically, matrix versus fracture domains. The second map simply does not contain any fracturing and therefore does not contribute directly to this particular discussion. However, we agree that providing it may offer useful context, and we will include this additional map in the supplement.

- **In lines 427-428: The sentence 'In the case of reliable samples, it is expected that the position of the spots does not influence the calculated ages.' What would you consider a reliable sample?**

Thank you for pointing this out. We agree that the definition of a "reliable sample" should be clarified. In the revised manuscript, we will specify that by "reliable sample," we refer to a sample with a homogeneous age and common Pb composition, sufficiently high U and Pb concentrations, and a wide enough range in isotopic ratios to allow for both accurate and precise U-Pb dating. We will revise the phrasing accordingly in the relevant section.

- **What if your sample isn't reliable, can you still use the approach?**

In our view, the proposed approach is particularly useful for assessing the reliability of a sample for geochronological purposes. As a first step, it is important to verify the consistency of the weighted mean ages obtained across the dataset. To support this, we propose to include a supplementary figure showing the weighted mean ages calculated for the different samples, based on 100 µm x 100 µm virtual spots in both the Tera-Wasserburg and Wetherill spaces. However, consistency alone is not sufficient to guarantee accuracy. As demonstrated by the C6-265-D5 example, coherent age estimates can still be biased. Therefore, as a second step, we believe it is essential to compare TW and Wetherill ages, which should agree within uncertainty. As discussed above, a discrepancy between these two regressions indicates potential bias and questions the reliability of the sample. We will expand the discussion on these points in the section addressing the limitations of our approach. Please also see below for additional details.

- **We would appreciate it if the authors could provide a bit more information on the limitations of this method. Particularly as carbonates can be complex in their formation and thereby age pattern. We would also appreciate it if the authors would touch on potential user biases that could be introduced when choosing one approach over another. It has been shown that data reduction and the choice of approach can make a difference in the final result. Particularly here with the possibility of choosing many virtual spots and statistical approaches, some clear guidelines for the reader would be helpful.**

In our virtual spot approach, we propose several strategies to obtain ages: the moving grid, sub-image, and Rectis methods. Among these, the moving grid is the primary and recommended approach and should always be used as the first step. We will clarify this point in the discussion. The sub-image and Rectis methods are complementary but optional. Specifically, the sub-image method presented here is more of a proof of concept, demonstrating the potential to date very small areas using our approach. Nonetheless, it can also contribute additional age constraints for larger samples through a weighted mean of sub-image results.

To help guide users through the most appropriate use of our method and to mitigate its main limitation—namely, the risk of biased ages for challenging

samples—we will add a workflow to the discussion. This will be presented both in text and as a graphic and will outline a logical sequence of steps:

1. Initial age calculation: Calculate a series of ages using 100 μm x 100 μm virtual spots in both TW and Wetherill spaces.

2. Check consistency: Assess (i) internal consistency of the ages in each space, and (ii) agreement between TW and Wetherill ages. Both conditions must be satisfied. If not, investigate the potential causes (e.g., low U/Pb concentrations, insufficient isotopic spread, multiple age populations). The datability of the sample should be questioned.

3. Age selection:
   (a) If the ages are consistent, select the most statistically robust result (e.g., lowest MSWD, highest precision), potentially adjusting the virtual spot size to improve the result.
   (b) If the TW and Wetherill ages are inconsistent, recalculate the ages after averaging the mass scans. If the TW and Wetherill results converge, return to step 3a. If not, the sample is likely undatable.

4. Optional steps:
   - Apply the Rectis method to obtain a potentially more precise age or one with better statistical parameters.
   - Use the sub-image method as an additional test. The weighted mean of the sub-image ages should ideally match the result obtained using the moving grid.

This workflow will be added to the revised manuscript to help users apply our method more effectively and transparently.

- **A number of studies (e.g., van Elteren et al., 2018, Norris et al., 2021) have shown that aliasing and other artefacts can be created by LA-ICP-MS mapping. Have you observed any such effects, how are you mitigating such effects and how would this influence your ages?**

We have not encountered such issues, except for a slight smearing effect related to the washout time, which should have no significant impact on the accuracy of the obtained ages (see also our response to Reviewer 1). However, it is plausible that effects such as aliasing could have a more substantial influence on the calculated isotopic ratios and, consequently, on the ages derived using our approach. This remains to be tested and represents an avenue for future investigation.

**Technical Comments:**

- **Use of the word 'image' when referring to the laser map, in many cases even using 'image map'. Images imply that pixels are acquired simultaneously, which is not the case for LA-ICP-MS. Therefore, the word map should be used in this case. Please change this throughout the manuscript.**

OK. We will follow these recommendations.

**Line 55 & 62: use 238U/208Pb, we were wondering if this is a typo and should say 238U/206Pb.**

This is not a typo. The $^{238}$U/$^{208}$Pb ratio is used in the approach of Drost et al. (2018). See their paper for additional detail.

**Line 76/77: 'Both allowed to obtain highly spatially resolved image maps (25 µm rasters) and with a good analytical sensitivity.' What is good analytical sensitivity in this case, can you please quantify this?**

Taking NIST SRM 614 as an example, at 500 Hz, analytical conditions used give about 100 kcps – 200 ckps / ppm $^{238}$U depending on the session. These values will be added to the text. Limits of detection and quantification are provided l. 147-148.

- **Line 169: 'This script as well as the ones described below are publicly available (Hoareau et al., 2024).' Instead of saying it is publicly available and referring to another paper it should say where the code is available. Good practice is to publish the code on GitHub and then provide the link here. Make it easy for people to find and use!**

We will ensure that the code made available on Zenodo is cited appropriately, as indicated in the Data Availability section. At this stage, we do not plan to publish the code on GitHub.

- **Line 172 should say virtual spot**

Yes.

- **Inconsistent use of NIST glass names, sometimes NIST SRM 612/614 other times NIST 612/614. We suggest to always use the same labelling.**

OK. We will use NIST SRM 612 and NIST SRM 614.

- **The use of the word spot height is a bit confusing, we suggest using width instead when talking about the vertical extent and then**

> **speaking about line length when speaking about the horizontal extent.**

We had the same remark from first reviewer. It will be changed.

- > **Line 183-185: "To achieve this, it may be necessary to adjust the size of the virtual spots very slightly (e.g., 51 μm x 50 μm instead of 50 μm x 50 μm) to ensure an integer number of pixels per spot."**
- > **Why is it, is it because of the Python language/computing characteristics?**

The user interface allows the user to specify virtual spot sizes in microns. The calculation then divides this length by the chosen mass scan duration to determine the number of pixels, which in turn defines the tolerated offsets between successive moving grids. Slightly adjusting the virtual spot size can result in a greater number of possible grid positions, as it leads to more integer values in the calculations. This behaviour is independent of the programming language used.

- > **Fig 1. The use of the arrows from A to B&C is confusing as they are not derived from A. They simply show a different configuration or the grid. Therefore, we suggest deleting the arrows. Furthermore, it would be good to provide the information on what map is shown. Is that a raw map? What element/ratio does it show? What sample is it?**

We agree. We will remove the arrows and add the ratio displayed ($^{238}$U/$^{206}$Pb). Sample is BH14 as explained in the caption.

- > **Fig 2. This figure is quite busy, are all images needed? Could some of them go into the supplement?**

Since the Figure presents the maps used and discussed in the study, we feel that keeping them in the main text is a minimum.

- > **In the caption of Fig 2, it says that concentrations were estimated from NIST SRM. First of all, which NIST was used for which sample? And secondly, what do you mean by estimates? Why not do a proper calibration of them?**

Concentrations are semi-quantitative estimates based on analysis of NIST SRM 612 (before 2020) and NIST SRM 614 (after 2020), as calculated from Iolite4. Purely quantitative concentrations would require the analysis either of $^{42}$Ca mass or that of a carbonate standard of homogenous U, Th and/or Pb concentration. The term 'estimates' can be replaced by 'semi-quantitative'.

- **Fig. 7: The maps at the top have two areas highlighted that represent two calcite generations. What is the rest of the map? Nowhere does it say what it is and based on what the areas highlighted have been chosen.**

The rest of the map is made of quartz (in blue) and calcite (in red). The highlighted areas have been chosen based in a sufficiently high $^{238}U/^{206}Pb$ for calcite rhombs (area 1) and the presence of a calcite-filled fracture (area 2). We will label the quartz grains and non-analysed calcite, as well as provide more detail in the text.

- **All isotope maps would benefit from a scale bar**

Right. We need to add them.

- **The labelling of the elements and isotope ratios next to the maps in the figures is very small and not well-readable. Suggest increasing the font size.**

OK.

**This is a co-review from Barbara Kunz & Igor Figueiredo**